# Gradient Flossing: Improving Gradient Descent through Dynamic Control of Jacobians

**Rainer Engelken**
Zuckerman Mind Brain Behavior Institute
Columbia University
New York, USA
`re2365@columbia.edu`

## Abstract

Training recurrent neural networks (RNNs) remains a challenge due to the instability of gradients across long time horizons, which can lead to exploding and vanishing gradients. Recent research has linked these problems to the values of Lyapunov exponents for the forward-dynamics, which describe the growth or shrinkage of infinitesimal perturbations. Here, we propose *gradient flossing*, a novel approach to tackling gradient instability by pushing Lyapunov exponents of the forward dynamics toward zero during learning. We achieve this by regularizing Lyapunov exponents through backpropagation using differentiable linear algebra. This enables us to "floss" the gradients, stabilizing them and thus improving network training. We demonstrate that *gradient flossing* controls not only the gradient norm but also the condition number of the long-term Jacobian, facilitating multidimensional error feedback propagation. We find that applying *gradient flossing* prior to training enhances both the success rate and convergence speed for tasks involving long time horizons. For challenging tasks, we show that *gradient flossing* during training can further increase the time horizon that can be bridged by backpropagation through time. Moreover, we demonstrate the effectiveness of our approach on various RNN architectures and tasks of variable temporal complexity. Additionally, we provide a simple implementation of our *gradient flossing* algorithm that can be used in practice. Our results indicate that *gradient flossing* via regularizing Lyapunov exponents can significantly enhance the effectiveness of RNN training and mitigate the exploding and vanishing gradients problem.

## 1 Introduction

Recurrent neural networks are commonly used both in machine learning and computational neuroscience for tasks that involve input-to-output mappings over sequences and dynamic trajectories. Training is often achieved through gradient descent by the backpropagation of error information across time steps [1, 2, 3, 4]. This amounts to unrolling the network dynamics in time and recursively applying the chain rule to calculate the gradient of the loss with respect to the network parameters. Mathematically, evaluating the product of Jacobians of the recurrent state update describes how error signals travel across time steps. When trained on tasks that have long-range temporal dependencies, recurrent neural networks are prone to exploding and vanishing gradients [5, 6, 7, 8]. These arise from the exponential amplification or attenuation of recursive derivatives of recurrent network states over many time steps. Intuitively, to evaluate how an output error depends on a small parameter change at a much earlier point in time, the error information has to be propagated through the recurrent network states iteratively. Mathematically, this corresponds to a product of Jacobians that describe how changes in one recurrent network state depend on changes in the previous network state. Together, this product forms the long-term Jacobian. The singular value spectrum of the long-term Jacobian reg-

37th Conference on Neural Information Processing Systems (NeurIPS 2023).

ulates how well error signals can propagate backwards along multiple time steps, allowing temporal credit assignment. A close mathematical correspondence of these singular values and the Lyapunov exponents of the forward dynamics was established recently [9, 10, 11, 12]. Lyapunov exponents characterize the asymptotic average rate of exponential divergence or convergence of nearby initial conditions and are a cornerstone of dynamical systems theory [13, 14]. We will use this link to improve the trainability of RNNs.

Previous approaches that tackled the problem of exploding or vanishing gradients have suggested solutions at different levels. First, specialized units such as LSTM and GRU were introduced, which have additional latent variables that can be decoupled from the recurrent network states via multiplicative (gating) interactions. The gating interactions shield the latent memory state, which can therefore transport information across multiple time steps [5, 6, 15]. Second, exploding gradients can be avoided by gradient clipping, which re-scales the gradient norm [16] or their individual elements [17] if they become too large [18]. Third, normalization schemes like batch normalization, layer norm and group norm prevent saturated nonlinearities that contribute to vanishing gradients [19, 20, 21]. Fourth, it was suggested that the problem of exploding/vanishing gradients can be ameliorated by specialized network architectures, for example, antisymmetric networks [22], orthogonal/unitary initializations [23, 24, 25], coupled oscillatory RNNs [26], Lipschitz RNNs [27], structured state space models [28, 29, 30, 31], echo state networks [32, 33], (recurrent) highway networks [34, 35], and stable limit cycle neural networks [11, 36, 37]. Fifth, for large networks, a suitable choice of weights can guarantee a well-conditioned Jacobian at initialization [23, 38, 39, 40, 41, 42, 43, 44, 45, 46, 47]. These initializations are based on mean-field methods, which become exact only in the large-network limit. Such initialization schemes have also been suggested for gated networks [45]. However, even when initializing the network with well-behaved gradients, gradients will typically not retain their stability during training once the network parameters have changed.

Here, we propose a novel approach to tackling this challenge by introducing *gradient flossing*, a technique that keeps gradients well-behaved throughout training. *Gradient flossing* is based on a recently described link between the gradients of backpropagation through time and Lyapunov exponents, which are the time-averaged logarithms of the singular values of the long-term Jacobian [9, 11, 12, 37]. *Gradient flossing* regularizes one or several Lyapunov exponents to keep them close to zero during training. This improves not only the error gradient norm but also the condition number of the long-term Jacobian. As a result, error signals can be propagated back over longer time horizons. We first demonstrate that the Lyapunov exponents can be controlled during training by including an additional loss term. We then demonstrate that *gradient flossing* improves the gradient norm and effective dimension of the gradient signal. We find empirically that *gradient flossing* improves test accuracy and convergence speed on synthetic tasks over a range of temporal complexities. Finally, we find that *gradient flossing* during training further helps to bridge long-time horizons and show that it combines well with other approaches to ameliorate exploding and vanishing gradients, such as dynamic mean-field theory for initialization, orthogonal initialization and gated units.
Our contributions include:

- *Gradient flossing*, a novel approach to the problem of exploding and vanishing gradients in recurrent neural networks based on regularization of Lyapunov exponents. [1]

- Analytical estimates of the condition number of the long-term Jacobian based on Lyapunov exponents.

- Empirical evidence that *gradient flossing* improves training on tasks that involve bridging long time horizons.

## 2   RNN Gradients and Lyapunov Exponents

We begin by revisiting the established mathematical relationship between the gradients of the loss function, computed via backpropagation through time, and Lyapunov exponents [9, 12], and how it relates to the problem of vanishing and exploding gradients. In backpropagation through time, network parameters $\theta$ are iteratively updated by stochastic gradient descent such that a loss $L_t$ is locally reduced [1, 2, 3, 4]. For RNN dynamics $\mathbf{h}_{s+1} = \mathbf{f}_\theta(\mathbf{h}_s, \mathbf{x}_{s+1})$, with recurrent network state $\mathbf{h}$, external input $\mathbf{x}$, and parameters $\theta$, the gradient of the loss $\mathcal{L}_t$ with respect to $\theta$ is evaluated by

---

[1]Code in Julia using Flux [77, 78] is available at https://github.com/RainerEngelken/GradientFlossing

unrolling the network dynamics in time. The resulting expression for the gradient is given by:

$$\frac{\partial \mathcal{L}_t}{\partial \theta} = \frac{\partial \mathcal{L}_t}{\partial \mathbf{h}_t} \sum_{\tau=t-l}^{\tau=t-1} \left( \prod_{\tau'=\tau}^{t-1} \frac{\partial \mathbf{h}_{\tau'+1}}{\partial \mathbf{h}_{\tau'}} \right) \frac{\partial \mathbf{h}_\tau}{\partial \theta} = \frac{\partial \mathcal{L}_t}{\partial \mathbf{h}_t} \sum_{\tau} \mathbf{T}_t(\mathbf{h}_\tau) \frac{\partial \mathbf{h}_\tau}{\partial \theta} \tag{1}$$

where $\mathbf{T}_t(\mathbf{h}_\tau)$ is composed of a product of one-step Jacobians $\mathbf{D}_s = \frac{\partial \mathbf{h}_{s+1}}{\partial \mathbf{h}_s}$:

$$\mathbf{T}_t(\mathbf{h}_\tau) = \prod_{\tau'=\tau}^{t-1} \frac{\partial \mathbf{h}_{\tau'+1}}{\partial \mathbf{h}_{\tau'}} = \prod_{\tau'=\tau}^{t-1} \mathbf{D}_{\tau'} \tag{2}$$

Due to the chain of matrix multiplications in $\mathbf{T}_t$, the gradients tend to vanish or explode exponentially with time. This complicates training particularly when the task loss at time $t$ dependents on inputs $\mathbf{x}$ or states $\mathbf{h}$ from many time steps prior which creates long temporal dependencies [5, 6, 7, 8]. How well error signals can propagate back in time is constrained by the tangent space dynamics along trajectory $\mathbf{h}_t$, which dictate how local perturbations around each point on the trajectory stretch, rotate, shear, or compress as the system evolves.

The singular values of the Jacobian's product $\mathbf{T}_t$, which determine how quickly gradients vanish or explode during backpropagation through time, are directly related to the Lyapunov exponents of the forward dynamics [9, 12]: Lyapunov exponents $\lambda_1 \geq \lambda_2 \cdots \geq \lambda_N$ are defined as the asymptotic time-averaged logarithms of the singular values of the long-term Jacobian [13, 48, 49]

$$\lambda_i = \lim_{t \to \infty} \frac{1}{t - \tau} \log(\sigma_{i,t}) \tag{3}$$

where $\sigma_{i,t}$ denotes the $i$th singular value of $\mathbf{T}_t(\mathbf{h}_\tau)$ with $\sigma_{1,t} \geq \sigma_{2,t} \ldots \sigma_{N,t}$ (See Appendix I for details). This means that positive Lyapunov exponents in the forward dynamics correspond to exponentially exploding gradient modes, while negative Lyapunov exponents in the forward dynamics correspond to exponentially vanishing gradient modes.

In summary, the Lyapunov exponents give the average asymptotic exponential growth rates of infinitesimal perturbations in the tangent space of the forward dynamics, which also constrain the signal propagation in backpropagation for long time horizons. Lyapunov exponents close to zero in the forward dynamics correspond to tangent space directions along which error signals are neither drastically attenuated nor amplified in backpropagation through time. Such close-to-neutral modes in the tangent dynamics can propagate information reliably across many time steps.

## 3   Gradient Flossing: Idea and Algorithm

We now leverage the mathematical connection established between Lyapunov exponents and the prevalent issue of exploding and vanishing gradients for regularizing the singular values of the long-term Jacobian. We term this procedure *gradient flossing*. To prevent exploding and vanishing gradients, we constrain Lyapunov exponents to be close to zero. This ensures that the corresponding directions in tangent space grow and shrink on average only slowly. This leads to a better-conditioned long-term Jacobian $\mathbf{T}_t(\mathbf{h}_\tau)$. We achieve this by using the sum of the squares of the first $k$ largest Lyapunov exponent $\lambda_1, \lambda_2 \ldots \lambda_k$ as a loss function:

$$\mathcal{L}_{\text{flossing}} = \sum_{i=1}^{k} \lambda_i^2 \tag{4}$$

and evaluate the gradient obtained from backpropagation through time:

$$\frac{\partial \mathcal{L}_{\text{flossing}}}{\partial \theta} = \sum_{i=1}^{k} \frac{\partial \lambda_i^2}{\partial \theta} \tag{5}$$

This might seem like an ill-fated enterprise, as the gradient expression in Eq 5 suffers from its own problem of exploding and vanishing gradients. However, instead of calculating the Lyapunov exponents by directly evaluating the long-term Jacobian $\mathbf{T}_t$ (Eq 2), we use an established iterative reorthonormalization method involving QR decomposition that avoids directly evaluating the ill-conditioned long-term Jacobian [12, 50].

First, we evolve an initially orthonormal system $\mathbf{Q}_s = [\mathbf{q}_s^1, \mathbf{q}_s^2, \dots \mathbf{q}_s^k]$ in the tangent space along the trajectory using the Jacobian $\mathbf{D}_s = \frac{\partial \mathbf{h}_{s+1}}{\partial \mathbf{h}_s}$. This means to calculate

$$\widetilde{\mathbf{Q}}_{s+1} = \mathbf{D}_s \mathbf{Q}_s \tag{6}$$

at every time-step. Second, we extract the exponential growth rates using the QR decomposition,

$$\widetilde{\mathbf{Q}}_{s+1} = \mathbf{Q}_{s+1} \mathbf{R}^{s+1},$$

which decomposes $\widetilde{\mathbf{Q}}_{s+1}$ uniquely into the product of an orthonormal matrix $\mathbf{Q}_{s+1}$ of size $N \times k$ so $\mathbf{Q}_{s+1}^\top \mathbf{Q}_{s+1} = \mathbb{1}_{k \times k}$ and an upper triangular matrix $\mathbf{R}^{s+1}$ of size $k \times k$ with positive diagonal elements. Note that the QR decomposition does not have to be applied at every step, just sufficiently often, i.e., once every $t_{\text{ONS}}$ such that $\widetilde{\mathbf{Q}}$ does not become ill-conditioned.

The Lyapunov exponents are given by time-averaged logarithms of the diagonal entries of $\mathbf{R}^s$ [49, 50]:

$$\lambda_i = \lim_{t \to \infty} \frac{1}{t} \log \prod_{s=1}^t \mathbf{R}_{ii}^s = \lim_{t \to \infty} \frac{1}{t} \sum_{s=1}^t \log \mathbf{R}_{ii}^s. \tag{7}$$

This way, the Lyapunov exponent can be expressed in terms of a temporal average over the diagonal elements of the $\mathbf{R}^s$-matrix of a QR decomposition of the iterated Jacobian. To propagate the gradient of the square of the Lyapunov exponents backward through time in *gradient flossing*, we used an analytical expression for the pullback of the QR decomposition [51]: The backward pass of the QR decomposition is given by [51, 52, 53, 54]

$$\overline{\mathbf{Q}} = \left[\overline{\mathbf{Q}} + \mathbf{Q} \, \texttt{copyltu}(\mathbf{M})\right] \mathbf{R}^{-T}, \tag{8}$$

where $\mathbf{M} = \mathbf{R}\overline{\mathbf{R}}^T - \overline{\mathbf{Q}}^T \mathbf{Q}$ and the $\texttt{copyltu}$ function generates a symmetric matrix by copying the lower triangle of the input matrix to its upper triangle, with the element $[\texttt{copyltu}(M)]_{ij} = M_{\max(i,j),\min(i,j)}$ [51, 52, 53, 54]. We denote here *adjoint variable* as $\overline{T} = \partial \mathcal{L} / \partial T$. A simple implementation of this algorithm in pseudocode is:

---
**Algorithm 1** Algorithm for *gradient flossing* of $k$ **tangent space directions**

---
   initialize $\mathbf{h}$, $\mathbf{Q}$
  **for** $e = 1 \to E$ **do**
     **for** $t = 1 \to T$ **do**
        $\mathbf{h} \leftarrow \mathbf{f}_\theta(\mathbf{h}, \mathbf{x})$
        $\mathbf{D} \leftarrow \frac{\mathrm{d}\mathbf{h}_t}{\mathrm{d}\mathbf{h}_{t-1}}$
        $\mathbf{Q} \leftarrow \mathbf{D} \cdot \mathbf{Q}$
        **if** $t \equiv 0 \pmod{t_{\text{ONS}}}$ **then**
           $\mathbf{Q}, \mathbf{R} \leftarrow \mathrm{qr}(\mathbf{Q})$
           $\gamma_i \mathrel{+}= \log(R_{ii})$
        **end if**
     **end for**
     $\lambda_i = \gamma_i / T$
     $\theta_{e+1} \leftarrow \theta_e - \eta \frac{\partial \mathcal{L}_{\text{flossing}}}{\partial \theta}$
  **end for**

---

For clarity, we described *gradient flossing* in terms of stochastic gradient descent, but we actually implemented it with the ADAM optimizer using standard hyperparameters $\eta$, $\beta_1$ and $\beta_2$. An example implementation in Julia [77] using Flux [78] is available here . Note that this algorithm also works for different recurrent network architectures. In this case, the Jacobians $\mathbf{D}$ has size $n \times n$, where $n$ is the number of dynamic variables of the recurrent network model. For example, in case of a single recurrent network of $N$ LSTM units, the Jacobian has size $2N \times 2N$ [9, 12, 46]. The Jacobian matrix $\mathbf{D}$ can either be calculated analytically or it can be obtained via automatic differentiation.

## 4 Gradient Flossing: Control of Lyapunov Exponents

In Fig 1, we demonstrate that *gradient flossing* can set one or several Lyapunov exponents to a target value via gradient descent with the ADAM optimizer in random Vanilla RNNs initialized with different weight variances. The $N$ units of the recurrent neural network follow the dynamics

$$\mathbf{h}_{s+1} = \mathbf{f}(\mathbf{h}_s, \mathbf{x}_{s+1}) = \mathbf{W}\phi(\mathbf{h}_s) + \mathbf{V}\mathbf{x}_{s+1}. \tag{9}$$

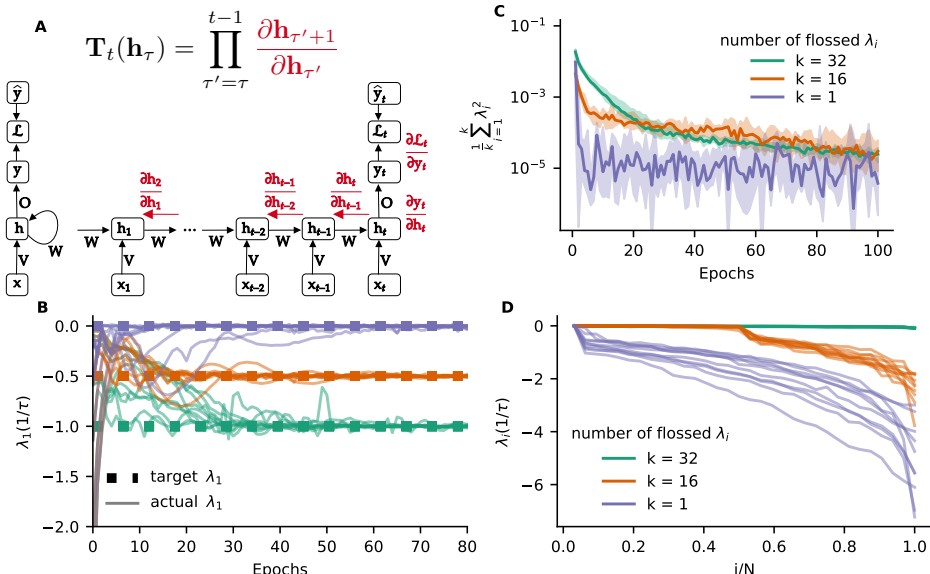

Figure 1: *Gradient flossing* **controls Lyapunov exponents and gradient signal propagation**
**A)** Exploding and vanishing gradients in backpropagation through time arise from amplifica-tion/attenuation of product of Jacobians that form the long-term Jacobian $\mathbf{T}_t(\mathbf{h}_\tau) = \prod_{\tau'=\tau}^{t-1} \frac{\partial \mathbf{h}_{\tau'+1}}{\partial \mathbf{h}_{\tau'}}$.
**B)** First Lyapunov exponent of Vanilla RNN as a function of training epochs. Minimizing the mean squared error between estimated first Lyapunov exponent and target Lyapunov exponent $\lambda_1 = -1, -0.5, 0$ by gradient descent. 10 Vanilla RNNs were initialized with Gaussian recurrent weights $W_{ij} \sim \mathcal{N}(0, g^2/N)$ where values of $g$ were drawn $g \sim \text{Unif}(0, 1)$. **C)** *Gradient flossing* minimizes the square of Lyapunov exponents over epochs. **D)** Full Lyapunov spectrum of Vanilla RNN after a different number of Lyapunov exponents are pushed to zero via *gradient flossing*. Note, the variability of the Lyapunov exponents that were not flossed. Parameters: network size $N = 32$ with 10 network realizations. Error bars in **C** indicate the 25% and 75% percentiles and solid line shows median.

The initial entries of $\mathbf{W}$ are drawn independently from a Gaussian distribution with zero mean and variance $g^2/N$, where $g$ is a gain parameter that controls the heterogeneity of weights. We here use the transfer function $\phi(x) = \tanh(x)$. (See appendix B for *gradient flossing* with ReLU and LSTM units). $\mathbf{x}_s$ is a sequence of inputs and $\mathbf{V}$ is the input weight. $\mathbf{x}_s$ is a stream of i.i.d. Gaussian input $x_s \sim \mathcal{N}(0, 1)$ and the input weights $\mathbf{V}$ are $\mathcal{N}(0, 1)$. Both $\mathbf{W}$ and $\mathbf{V}$ are trained during *gradient flossing*.

In Fig 1B, we show that for randomly initialized RNNs, the Lyapunov exponent can be modified by *gradient flossing* to match a desired target value. The networks were initialized with 10 different values of initial weight strength $g$ chosen uniformly between 0 and 1. During *gradient flossing*, they quickly approached three different target values of the first Lyapunov exponents $\lambda_1^{\text{target}} = \{-1, -0.5, 0\}$ within less than 100 training epochs with batch size $B = 1$. We note that *gradient flossing* with positive target $\lambda_1^{\text{target}}$ seems not to arrive at a positive Lyapunov exponent $\lambda_1$.

Fig 1C shows *gradient flossing* for different numbers of Lyapunov exponents $k$. Here, during gradient-descent, the sum of the squares of 1, 16, or 32 Lyapunov exponents is used as loss in *gradient flossing* (see Fig 1A). Fig 1D shows the Lyapunov spectrum after flossing, which now has 1, 16, or 32 Lyapunov exponents close to zero. We conclude that *gradient flossing* can selectively manipulate one, several, or all Lyapunov exponents before or during network training. *Gradient flossing* also works for RNNs of ReLU and LSTM units (See appendix B. Further, we find that the computational bottleneck of *gradient flossing* is the QR decomposition, which has a computational complexity of $\mathcal{O}\left(N\,k^2\right)$, both in the forward pass and in the backward pass. Thus, *gradient flossing* of the entire Lyapunov spectrum is computationally expensive. However, as we will show, not all Lyapunov exponents need to be flossed and only short episodes of *gradient flossing* are sufficient for significantly improving the training performance.

# 5 Gradient Flossing: Condition Number of the Long-Term Jacobian

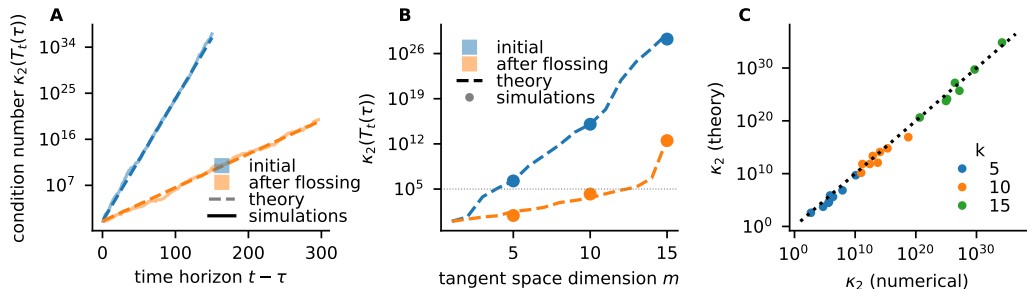

Figure 2: ***Gradient flossing* reduces condition number of the long-term Jacobian A)** Condition number $\kappa_2$ of long-term Jacobian $\mathbf{T}_t(\mathbf{h}_\tau)$ as a function of time horizon $t - \tau$ at initialization (blue) and after *gradient flossing* (orange). Direct numerical simulations are done with arbitrary precision floating point arithmetic (transparent lines) with 256 bits per float, asymptotic theory based on Lyapunov exponents (dashed lines) (Eq 10). **B)** Condition number for different number of tangent space dimensions $m$. Simulations (dots) and Lyapunov exponent based theory (dashed lines) at initialization (blue) and after *gradient flossing* (orange). *Gradient flossing* increases the number of tangent space dimensions available for backpropagation for a given condition number (Grey dotted line as a guide for eye for $\kappa_2 = 10^5$.) First 15 Lyapunov exponents were flossed. **C)** Comparison of condition number obtained via direct numerical simulations vs. Lyapunov exponent-based. Colors denote the number of flossed Lyapunov exponents $k$. Parameters: $g = 1$, batch size $b = 1$, $N = 80$, epochs $= 500$, $T = 500$, *gradient flossing* for $E_f = 500$ epochs. Input $\mathbf{x}_s$ identical to delayed XOR task in Fig 3D.

A well-conditioned Jacobian is essential for efficient and fast learning [23, 55, 56]. *Gradient flossing* improves the condition number of the long-term Jacobian which constrains the error signal propagation across long time horizons in backpropagation (Fig 2). The condition number $\kappa_2$ of a linear map $A$ measures how close the map is to being singular and is given by the ratio of the largest singular value $\sigma_{\max}$ and the smallest singular values $\sigma_{\min}$, so $\kappa_2(A) = \frac{\sigma_{\max}(A)}{\sigma_{\min}(A)}$. According to the rule of thumb given in [57], if $\kappa_2(A) = 10^p$, one can anticipate losing at least $p$ digits of precision when solving the equation $Ax = b$. Note that the long-term Jacobian $\mathbf{T}_t$ is composed of a product of Jacobians, which generically makes it ill-conditioned. To nevertheless quantify the condition number numerically, we use arbitrary-precision arithmetic with 256 bits per float. We find numerically that the condition number of $\mathbf{T}_t$ exponentially diverges with the number of time steps (Fig 2A). We compare the numerically measured condition number $\kappa_2$ with an asymptotic approximation of the condition number based on Lyapunov exponents that are calculated in the forward pass and find a good match (Fig 2A).

Our theoretical estimate of the condition number $\kappa_2$ of an orthonormal system $\mathbf{Q}$ of size $N \times m$ that is temporally evolved by the long-term Jacobian $\mathbf{T}_t$ is:

$$\kappa_2(\widetilde{\mathbf{Q}}_{t+\tau}) = \kappa_2\big(\mathbf{T}_t(\mathbf{h}_\tau)\mathbf{Q}_t\big) = \frac{\sigma_1(\mathbf{T}_t(\mathbf{h}_\tau))}{\sigma_m(\mathbf{T}_t(\mathbf{h}_\tau))} \approx \exp\big((\lambda_1 - \lambda_m)(t - \tau)\big). \tag{10}$$

where $\sigma_1(\mathbf{T}_t(\mathbf{h}_\tau))$ and $\sigma_m(\mathbf{T}_t(\mathbf{h}_\tau))$ are the first and $m$th singular value of the long-term Jacobian. We note that this theoretical estimate of the condition number follows from the asymptotic definition of Lyapunov exponents and should be exact in the limit of long times. We find that *gradient flossing* reduces the condition number by a factor whose magnitude increases exponentially with time (orange in Fig 2A). Thus, we can expect that *gradient flossing* has a stronger effect on problems with a long time horizon to bridge. We will later confirm this numerically.

Moreover, Lyapunov exponents enable the estimation of the number of gradient dimensions available for the backpropagation of error signals. Generally, the long-term Jacobian is ill-conditioned, however, the Lyapunov spectrum provides for a given number of tangent space dimensions an estimate of the condition number. This indicates how close to singular the gradient signal for a given number of tangent space dimensions is. Given a fixed acceptable condition number—determined, for example, by noise level or floating-point precision—we observe that *gradient flossing* increases the number of usable tangent space dimensions for backpropagation (Fig 2B).

Finally, we show that the asymptotic estimate of the condition number based on Lyapunov exponents can even predict differences in condition number that originate from finite network size $N$ (Fig 2C). We emphasize that this goes beyond mean-field methods, which become exact only in the large-network limit $N \to \infty$ and usually do not capture finite-size effects [58] (see appendix G).

# 6 Initial *Gradient Flossing* Improves Trainability

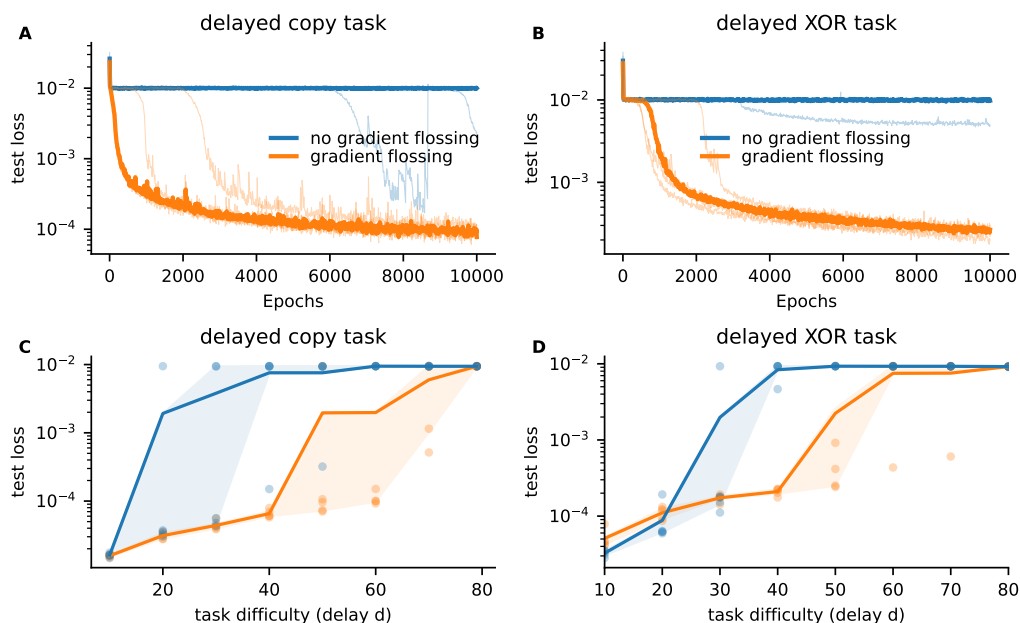

Figure 3: ***Gradient flossing* improves trainability on tasks that involve long time horizons A)** Test error for Vanilla RNNs trained on delayed copy task $y_t = x_{t-d}$ for $d = 40$ with and without *gradient flossing* flossing. Solid lines are medians across 5 network realizations. **B)** Same as **A** for delayed XOR task with $y_t = |x_{t-d/2} - x_{t-d}|$. **C)** Mean final test loss as a function of task difficulty (delay $d$) for delayed copy task. **D)** Mean final test loss as a function of task difficulty (delay $d$) for delayed XOR task. Parameters: $g = 1$, batch size $b = 16$, $N = 80$, epochs $= 10^4$, $T = 300$, *gradient flossing* for $E_f = 500$ epochs on $k = 75$ before training. Shaded regions in **C** and **D** indicate the 20% and 80% percentiles and solid line shows mean. Dots are individual runs. Task loss: $\text{MSE}(y, \hat{y})$.

We next present numerical results on two tasks with variable spatial and temporal complexity, demonstrating that *gradient flossing* before training improves the trainability of Vanilla RNNs. We call *gradient flossing* before training in the following preflossing. For preflossing, we first initialize the network randomly, then minimize $\mathcal{L}_{\text{flossing}} = \sum_{i=1}^{k} \lambda_i^2$ using the ADAM optimizer and subsequently train on the tasks. We deliberately do not use sequential MNIST or similar toy tasks commonly used to probe exploding/vanishing gradients, because we want a task where the structure of long-range dependencies in the data is transparent and can be varied as desired.

First, we consider the delayed copy task, where a scalar stream of random input numbers $x$ must be reproduced by the output $y$ delayed by $d$ time steps, i.e. $y_t = x_{t-d}$. Although the task itself is trivial and can be solved even by a linear network through a delay line (see appendix E), RNNs encounter vanishing gradients for large delays $d$ during training even with 'critical' initialization with $g = 1$. Our experiments show that *gradient flossing* can substantially improve the performance of RNNs on this task (Fig 3A, C). While Vanilla RNNs without *gradient flossing* fail to train reliably beyond $d = 20$, Vanilla RNNs with *gradient flossing* can be reliably trained for $d = 40$ (Fig 3C). Note that we flossed here $k = 40$ Lyapunov exponents before training. We will later investigate the role of the number of flossed Lyapunov exponents.

Second, we consider the temporal XOR task, which requires the RNN to perform a nonlinear input-output computation on a sequential stream of scalar inputs, i.e., $y_t = |x_{t-d/2} - x_{t-d}|$, where $d$ denotes a time delay of $d$ time steps (For details see appendix H). Fig 3D demonstrates that *gradient flossing* helps to train networks on a substantially longer delay $d$. We found similar improvements through *gradient flossing* for RNNs initialized with orthogonal weights (see appendix G).

# 7 *Gradient Flossing* During Training

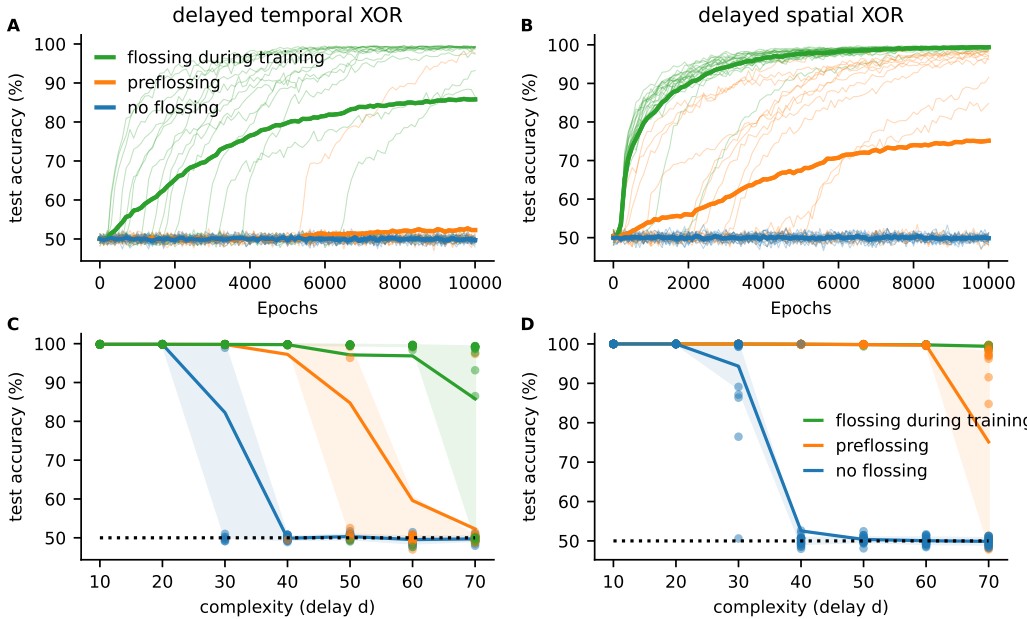

Figure 4: *Gradient flossing* **during training further improves trainability**
**A)** Test accuracy for Vanilla RNNs trained on delayed temporal binary XOR task $y_t = x_{t-d/2} \oplus x_{t-d}$ with *gradient flossing* during training (green), preflossing (*gradient flossing* before training) (orange), and with no *gradient flossing* (blue) for $d = 70$. Solid lines are mean across 20 network realizations, individual network realizations shown in transparent fine lines. **B)** Same as **A** for delayed spatial XOR task with $y_t = x_{t-d}^1 \oplus x_{t-d}^2 \oplus x_{t-d}^3$ . Parameters ($g = 1$, batch size $b = 16$). **C)** Test accuracy as a function of task difficulty (delay $d$) for delayed temporal XOR task. **D)** Test accuracy as a function of task difficulty (delay $d$) for delayed spatial XOR task. Parameters: $g = 1$, batch size $b = 16$, $N = 80$, epochs $= 10^4$, $T = 300$, *gradient flossing* for $E_f = 500$ epochs on $k = 75$ before training and during training for green lines, and only before training for orange lines. Same plotting conventions as previous figure. Task loss: cross-entropy between $y$ and $\hat{y}$.

We next investigate the effects of *gradient flossing* during the training and find that *gradient flossing* during training can further improve trainability. We trained RNNs on two more challenging tasks with variable temporal complexity and performed *gradient flossing* either both during and before training, only before training, or not at all.

Fig 4A shows the test accuracy for Vanilla RNNs training on the delayed temporal XOR task $y_t = x_{t-d/2} \oplus x_{t-d}$ with random Bernoulli process $x \in \{0, 1\}$. The accuracy of Vanilla RNNs falls to chance level for $d \geq 40$ (Fig 4C). With *gradient flossing* before training, the trainability can be improved, but still goes to chance level for $d = 70$. In contrast, for networks with *gradient flossing* during training, the accuracy is improved to $> 80\%$ at $d = 70$. In this case, we preflossed for 500 epochs before task training and again after 500 epochs of training on the task. In Fig 4B, D the networks have to perform the nonlinear XOR operation $y_t = x_{t-d}^1 \oplus x_{t-d}^2 \oplus x_{t-d}^3$ on a three-dimensional binary input signal $x^1$, $x^2$, and $x^3$ and generate the correct output with a delay of $d$ steps. While the solution of the task itself is not difficult and could even be implemented by hand (see appendix), the task is challenging for backpropagation through time because nonlinear temporal associations bridging long time horizons have to be formed. Again, we observe that *gradient flossing* before training improves the performance compared to baseline, but starts failing for long delays $d > 60$. In contrast, networks that are also flossed during training can solve even more difficult tasks (Fig 4D). We find that after *gradient flossing*, the norm of the error gradient with respect to initial conditions $\mathbf{h}_0$ is amplified (appendix C). Interestingly, *gradient flossing* can also be detrimental to task performance if it is continued throughout all training epochs (appendix C)

We note that merely regularizing the spectral radius of the recurrent weight matrix $\mathbf{W}$ or the individual one-step Jacobians $\mathbf{D}_s$ numerically or based on mean-field theory does not yield such a training improvement. This suggests that taking the temporal correlations between Jacobians $\mathbf{D}_s$ into account is important for improving trainability.

### 7.1 *Gradient Flossing* for Different Numbers of Flossed Lyapunov Exponents

We investigated how many Lyapunov exponents $k$ have to be flossed to achieve an improvement in training success (Fig 5). We studied this in the binary temporal delayed XOR task with *gradient flossing* during training (same as Fig 3) and varied the task difficulty by changing the delay $d$.

We found that as the task becomes more difficult, networks where not enough Lyapunov exponents $k$ are flossed begin to fall below 100% test accuracy (Fig 5A). Correspondingly, when measuring final test accuracy as a function of the number of flossed Lyapunov exponents, we observed that more Lyapunov exponent $k$ have to be flossed to achieve 100% accuracy as the tasks become more difficult (Fig 5B). We also show the entire parameter plane of median test accuracy as a function of both number of flossed Lyapunov exponents $k$ and task difficulty (delay $d$), and found the same trend (Fig 5B). Overall, we found that tasks with larger delay $d$ require more Lyapunov exponents close to zero. We note that this might also partially be caused by the 'streaming' nature of the task: in our tasks, longer delays automatically imply that more values have to be stored as at any moment all the values in the 'delay line' have to be remembered to successfully solve the tasks. This is different from tasks where a single variable has to be stored and recalled after a long delay. It would be interesting to study tasks where the number of delay steps and the number of items in memory can be varied independently.

Finally, we did the same analysis on networks with only preflossing (*gradient flossing* before training) and found the same trend (supplement Fig 7D), however, in that case even if all $N$ Lyapunov exponents were flossed, thus $k = N$, they were not able to solve the most difficult tasks. This seems to indicate that *gradient flossing* during training cannot be replaced by just *gradient flossing* more Lyapunov exponents before training.

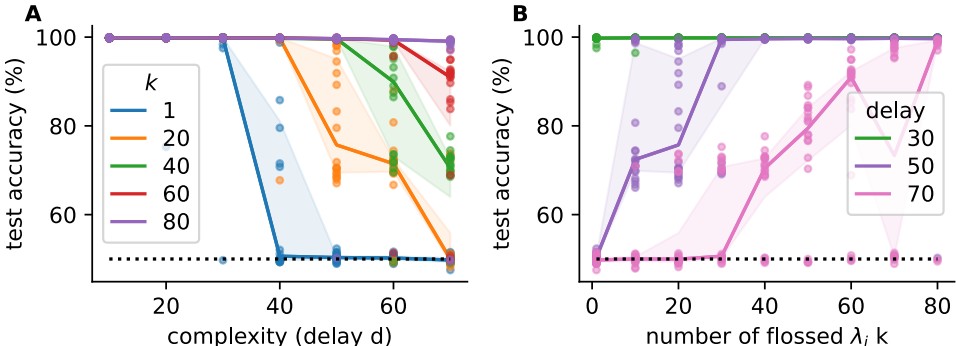

**Figure 5:** *Gradient flossing* **for different numbers of flossed Lyapunov exponents**
**A)** Test accuracy for delayed temporal XOR task as a function of delay $d$ with different numbers flossed Lyapunov exponents $k$. **B)** Same data as **A** but here test accuracy as a function of number of flossed Lyapunov exponents $k$. Parameters: $g = 1$, batch size $b = 16$, $N = 80$, epochs $= 10^4$ for delayed temporal XOR, epochs $= 5000$ for delayed spatial XOR, $T = 300$, *gradient flossing* for $E_f = 500$ epochs before training and during training for **A, B**. Shaded areas are 25% and 75% percentile, solid lines are means, transparent dots are individual simulations, task loss: cross-entropy between $y$ and $\hat{y}$.

## 8   Limitations

The mathematical connection between Lyapunov exponents and backpropagation through time exploited in *gradient flossing* is rigorously established only in the infinite-time limit. It would be interesting to extend our analysis to finite-time Lyapunov exponents.

Furthermore, the backpropagation through time gradient involves a sum over products of Jacobians of different time periods $t - \tau$, but the Lyapunov exponent only considers the asymptotic longest product. Additionally, Lyapunov exponents characterize the asymptotic dynamics on the attractor of the dynamics, whereas RNNs often exploit transient dynamics from some initial conditions outside or towards the attractor.

Although our proposed method focuses on exploiting Lyapunov exponents, it neglects the geometry of covariant Lyapunov vectors [59], which could be used to improve training performance, speed, and reliability. Additionally, it is important to investigate how sensitive the method is to the choice of orthonormal basis employed because it is only guaranteed to become unique asymptotically [60].

Finally, the computational cost of our method scales with $O(Nk^2)$, where $N$ is the network size and $k$ is the number of Lyapunov exponents calculated. To reduce the computational cost, we suggest doing QR decomposition only sufficiently often to ensure that the orthonormal system is not ill-conditioned and using *gradient flossing* only intermittently or as pretraining. One could also calculate the Lyapunov spectrum for a shorter time interval or use a cheaper proxy for the Lyapunov spectrum and investigate more efficient *gradient flossing* schedules.

## 9 Discussion

We tackle the problem of gradient signal propagation in recurrent neural networks through a dynamical systems lens. We introduce a novel method called *gradient flossing* that addresses the problem of gradient instability during training. Our approach enhances gradient signal stability both before and during training by regularizing Lyapunov exponents. By keeping the long-term Jacobian well-conditioned, *gradient flossing* optimizes both training accuracy and speed. To achieve this, we combine established dynamical systems methods for calculating Lyapunov exponents with an analytical pullback of the QR factorization. This allows us to establish and maintain gradient stability in a in a manner that is memory-efficient, numerically stable, and exact across long time horizons. Our method is applicable to arbitrary RNN architectures, nonlinearities, and also neural ODEs [61]. Empirically, pre-training with *gradient flossing* enhances both training speed and accuracy. For difficult temporal credit assignment problems, *gradient flossing* throughout training further enhances signal propagation. We also demonstrate the versatility of our method on a set of synthetic tasks with controllable time-complexity and show that it can be combined with other approaches to tackle exploding and vanishing gradients, such as dynamic mean-field theory for initialization, orthogonal initialization and specialized single units, such as LSTMs.

Prior research on exploding and vanishing gradients mainly focused on selecting network architectures that are less prone to exploding/vanishing gradients or finding parameter initializations that provide well-conditioned gradients at least at the beginning of training. Our introduced *gradient flossing* can be seen as a complementary approach that can further enhance gradient stability throughout training. Compared to the work on picking good parameter initializations based on random matrix theory [46] and mean-field heuristics [45], *gradient flossing* provides several improvements: First, mean-field theory only considers the gradient flow at initialization, while *gradient flossing* can maintain gradient flow and well-conditioned Jacobians throughout the training process. Second, random matrix theory and mean-field heuristics are usually confined to the limit of large networks [58], while *gradient flossing* can be used for networks of any size. The link between Lyapunov exponents and the gradients of backpropagation through time has been described previously [9, 12] and has been spelled out analytically and studied numerically [10, 11, 62, 63, 64]. In contrast, we use Lyapunov exponents here not only as a diagnostic tool for gradient stability but also to show that they can directly be part of the cure for exploding and vanishing gradients.

Future investigations could delve further into the roles of the second to $N$th Lyapunov exponents in trainability, and how it is related to the task at hand, the rank of the parameter update, the dimensionality of the solution space, as well as the network dynamics (see also [37, 65, 66]). Our results suggest a trade-off between trainability across long time horizons and the nonlinear task demands that is worth exploring in more detail (appendix C). Applying *gradient flossing* to real-time recurrent learning and its biologically plausible variants is another avenue [67]. Extending *gradient flossing* to feedforward networks, state-space models and transformers is a promising avenue for future research (see also [31, 47, 68, 69, 70, 71, 72]). While Lyapunov exponents are only strictly defined for dynamical systems, such as maps or flows that are endomorphisms, the long-term Jacobian of deep feedforward networks could be treated similarly. This could also provide a link between the stability of the network against adversarial examples and its dynamic stability, as measured by Lyapunov exponents. Given that time-varying input can suppress chaos in recurrent networks [9, 12, 73, 74, 75, 76], we anticipate they may exacerbate vanishing gradients. *Gradient flossing* could also be applied in neural architecture search, to identify and optimize trainable networks. Finally, *gradient flossing* is applicable to other model parameters, as well. For instance, gradients of Lyapunov exponents with respect to single-unit parameters could optimize the activation function and single-neuron biophysics in biologically plausible neuron models.

## Acknowledgments and Disclosure of Funding

I thank E. Izhikevich, F. Wolf, S. Goedeke, J. Lindner, L.F. Abbott, L. Logiaco, M. Schottdorf, G. Wayne and P. Sokol for fruitful discussions and J. Stone, L.F. Abbott, M. Ding, O. Marshall, S. Goedeke, S. Lippl, M.P. Puelma-Touzel, J. Lindner and the reviewers for feedback on the manuscript. I thank Jinguo Liu for the Julia package BackwardsLinalg.jl and also the developers of Julia and Flux [77, 78]. Research supported by NSF NeuroNex Award (DBI-1707398), the Gatsby Charitable Foundation (GAT3708), the Simons Collaboration for the Global Brain (542939SPI), and the Swartz Foundation (2021-6).

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

## A    Backpropagation Through QR Decomposition

The backward pass of the QR decomposition is given by [51, 52, 53, 54]

$$\overline{\mathbf{Q}} = \left[\overline{\mathbf{Q}} + \mathbf{Q}\,\mathtt{copyltu}(\mathbf{M})\right]\mathbf{R}^{-T} \tag{11}$$

where $\mathbf{M} = \mathbf{R}\overline{\mathbf{R}}^T - \overline{\mathbf{Q}}^T\mathbf{Q}$ and the $\mathtt{copyltu}$ function generates a symmetric matrix by copying the lower triangle of the input matrix to its upper triangle, with the element $[\mathtt{copyltu}(M)]_{ij} = M_{\max(i,j),\min(i,j)}$ [51, 52, 53, 54]. *Adjoint variable* are written here as $\overline{T} = \partial\mathcal{L}/\partial T$.

Using an analytical pullback is more memory-efficient and less computationally costly than directly doing automatic differentiation through the QR-decomposition. Moreover, from a practical perspective, for QR decomposition, often BLAS/LAPACK routines are utilized which are not amenable to common differentiable programming frameworks like TensorFlow, PyTorch, JAX and Zygote. In our implementation of *gradient flossing*, we used the Julia package BackwardsLinalg.jl by Jinguo Liu available at here .

## B    Further Details and Analysis of Gradient Flossing

An example implementation of *gradient flossing* in Flux [78], a machine learning library in Julia [77] is available at https://github.com/RainerEngelken/GradientFlossing. We are actively developing implementations for other widely used differentiable programming frameworks.

### B.1    *Gradient Flossing* for recurrent LSTM and ReLU networks

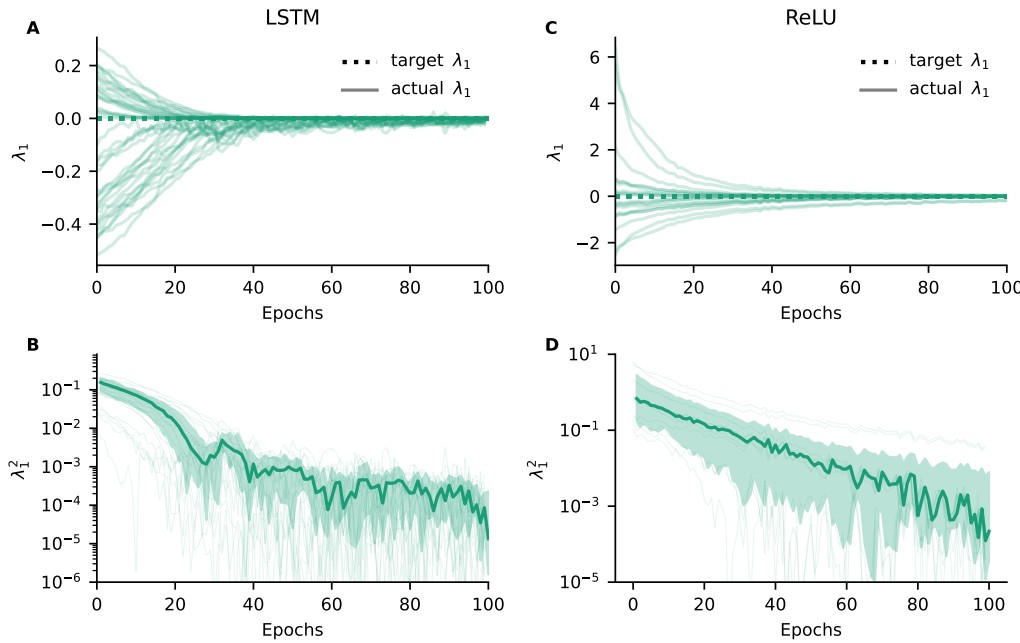

Figure 6: *Gradient flossing* **for recurrent LSTM networks and recurrent ReLU networks**
**A)** First Lyapunov exponent of LSTM network as a function of training epochs. Minimizing the mean squared error between estimated first Lyapunov exponent and target Lyapunov exponent $\lambda_1 = 0$ by gradient descent. First Lyapunov exponent of LSTM network (solid lines) converges to target value (thick dashed lines) within less than 100 epochs. 10 random LSTM RNNs were initialized with Gaussian recurrent weights, where standard deviations of weight scaling were drawn $g \sim \mathrm{Unif}(0,1)$. **B)** *Gradient flossing* minimizes the square of the first Lyapunov exponent of random recurrent LSTM networks over epochs. **C)** Same as **A** for recurrent ReLU network. Here networks were initialized with Gaussian recurrent weights $W_{ij} \sim \mathcal{N}(-0.1, g^2/N)$ where values of $g$ were drawn $g \sim \mathrm{Unif}(0,1)$ **D)** **B)** for recurrent ReLU network. Parameters: network size $N = 32$ with 10 network realizations. Shaded regions in **B**, **D** are 25% and 75% percentiles, solid line shows median.

We demonstrate that *gradient flossing* can also be applied to recurrent LSTM and ReLU networks in Fig 6. To this end, we generated random LSTM networks where the weights of all the different gates and biases were independently and identically distributed (i.i.d.) and sampled from Gaussian distributions of different variance. Our results show that *gradient flossing* can also constrain the Lyapunov exponent to be close to zero. The dynamics of each of the $N$ LSTM units follows the map [6]:

$$
\begin{aligned}
f_t &= \sigma_g(U_f h_{t-1} + W_f x_t + b_f) & (12) \\
o_t &= \sigma_g(U_o h_{t-1} + W_o x_t + b_o) & (13) \\
i_t &= \sigma_g(U_i h_{t-1} + W_i x_t + b_i) & (14) \\
\tilde{c}_t &= \sigma_h(U_c h_{t-1} + W_c x_t + b_c) & (15) \\
c_t &= f_t \odot c_{t-1} + i_t \odot \tilde{c}_t & (16) \\
h_t &= o_t \odot \phi(c_t) & (17)
\end{aligned}
$$

where $\odot$ denotes the Hadamard product, $\sigma_g(x) = \frac{1}{1+\exp(-x)}$ is the sigmoid function, $\sigma_h(x) = \tanh(x)$ and entries of the matrices $U_x$ are drawn from $U_x \sim \mathcal{N}(0, g_x^2/N)$. For simplicity, the bias terms $b_x$ are scalars. Subscripts $f$, $o$ and $i$ denote respectively the forget gate, the output gate, the input gate, and $c$ is the cell state. In each LSTM unit, there are two dynamic variables $c$ and $h$, and three gates $f$, $o$, and $i$ that control the flow if signals into and out of the cell $c$. We set the values $g_{ih}, g_{ix}, g_{fx}, b_f, g_{ch}, g_{cx}, g_{cx}, g_{ox}$ to be uniformly distributed between 0 and 1 and initialize $b_i, g_{fh}, b_c, b_0$ as zero.

During *gradient flossing*, the actual Lyapunov exponents of different random network realizations converge close to the target Lyapunov exponent $\lambda_1^{\text{target}} = 0$ in fewer than 100 epochs as shown in Fig 6A. Fig 6B shows that the squared Lyapunov exponents converge towards zero. We note that for LSTM networks, a target Lyapunov exponent of $\lambda_1^{\text{target}} = -1$ is achieved after 100 *gradient flossing* steps only for a subset of random network realizations (not shown). We speculate that behavior is influenced by the gating structure of LSTM units, which seems to naturally place the first Lyapunov exponent close to zero for certain initializations (See also [9, 12, 46, 64]).

For the recurrent ReLU networks, we considered the same Vanilla RNN dynamics as in the main manuscript in Eq 9

$$
\mathbf{h}_{s+1} = \mathbf{f}(\mathbf{h}_s, \mathbf{x}_{s+1}) = \mathbf{W}\phi(\mathbf{h}_s) + \mathbf{V}\mathbf{x}_{s+1},
$$

The initial entries of $\mathbf{W}$ are drawn independently from a Gaussian distribution with a negative mean of $-0.1$ and variance $g^2/N$, where $g$ is a gain parameter that controls the heterogeneity of weights. We use the transfer function $\phi(x) = \max(x, 0)$. $\mathbf{x}_s$ is a sequence of inputs and $\mathbf{V}$ is the input weight. $\mathbf{x}_s$ is a stream of i.i.d. Gaussian input $x_s \sim \mathcal{N}(0, 1)$ and the input weights $\mathbf{V}$ are $\mathcal{N}(0, 1)$. Both $\mathbf{W}$ and $\mathbf{V}$ are trained during *gradient flossing*. We found that some ReLU network had initially unstable dynamics with positive Lyapunov exponents Fig 6C. However, during *gradient flossing*, these unstable networks were quickly stabilized. Fig 6D shows that the squared Lyapunov exponents of ReLU networks converge towards zero.

## B.2   Additional Results for Different Numbers of Flossed Lyapunov Exponents

Additionally to the main Fig 5, we did the same analysis on networks with only preflossing (*gradient flossing* before training) and found that more Lyapunov exponent $k$ have to be flossed to achieve 100% accuracy as the tasks become more difficult (Fig 7D), however, in that case even if all $N$ Lyapunov exponents were flossed, thus $k = N$, they were not able to solve the most difficult tasks. This seems to indicate that *gradient flossing* during training cannot be replaced by just *gradient flossing* more Lyapunov exponents before training.

# C   Additional Results on *Gradient Flossing* Throughout Training

We now discuss some additional results on *gradient flossing* throughout training. First, we analyze how *gradient flossing* affects the gradients and find that during *gradient flossing*, the norm of gradients that bridge many time steps are boosted. Moreover, subordinate singular values of the error norm of the recurrent weights are also boosted, indicating that *gradient flossing* can increase the effective rank of the parameter update. Additionally, we show that if *gradient flossing* is continued throughout training it can be detrimental to the accuracy. Finally, we show that Lyapunov exponents of successfully trained networks after training for the spatial delayed XOR task have a simple relationship to the delay $d$.

# D   *Gradient Flossing* boosts the Gradient Norm for Long Time Horizons

In this section, we investigate the impact of *gradient flossing* on the norm and structure of the gradient. It is important to note that the complete error gradient of backpropagation through time is composed of a summation of products of one-step Jacobians, reflecting the number of "loops" the error signal traverses through the recurrent

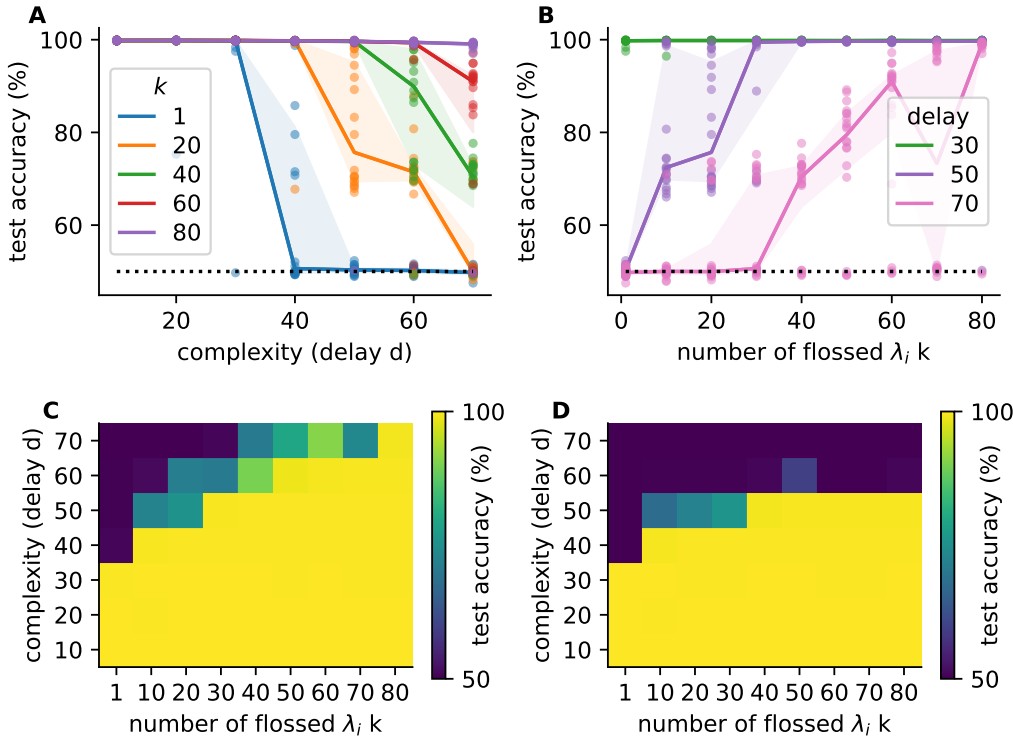

Figure 7: *Gradient flossing* **for different numbers of flossed Lyapunov exponents**
**A)** Test accuracy for delayed temporal XOR task as a function of delay $d$ with different numbers flossed Lyapunov exponents $k$. **B)** Same data as **A** but here test accuracy as a function of number of flossed Lyapunov exponents $k$. **C)** Median test accuracy for delayed temporal XOR task as a function of delay $d$ and $k$ for networks with *gradient flossing* during training (500 steps of *gradient flossing* at epochs $e \in \{0, 100, 200, 300, 400\}$). **D)**Same as **B** for *preflossing* only. Parameters: $g = 1$, batch size $b = 16$, $N = 80$, epochs $= 10^4$ for delayed temporal XOR, epochs $= 5000$ for delayed spatial XOR, $T = 300$, *gradient flossing* for $E_f = 500$ epochs before training and during training for **A, B, C**, and only before training for **C**. Shaded areas are 25% and 75% percentiles, solid lines are means, transparent dots are individual simulations, task loss: cross-entropy btw. $y, \hat{y}$.

dynamics before reaching its target. Consequently, when the singular values of the long-term Jacobian are smaller than 1, the influence of the shorter loops typically dominates the long-term Jacobian.

In our tasks, we have full control over the correlation structure of the task and thus know exactly which loop length of backpropagation through time is necessary for finding the correct association. We were moreover careful in our task design not to have any additional signals in our task that might help to bridge the long time scale. In the case of vanishing gradients, the gradient norm is predominantly influenced by the shorter loops, even though the actual signal in the gradient originates solely from the loop of length $d$ in our task. To mitigate the contamination of spurious signals from shorter loops and effectively extract the gradient that spans long time horizons, we focus on the gradient with respect to the initial conditions $\mathbf{h}_0$.

$$\frac{\partial \mathcal{L}_t}{\partial \mathbf{h}_0} = \frac{\partial \mathcal{L}_t}{\partial \mathbf{h}_t} \sum_{\tau=t-l}^{\tau=t-1} \left( \prod_{\tau'=\tau}^{t-1} \frac{\partial \mathbf{h}_{\tau'+1}}{\partial \mathbf{h}_{\tau'}} \right) \frac{\partial \mathbf{h}_\tau}{\partial \mathbf{h}_0} = \frac{\partial \mathcal{L}_t}{\partial \mathbf{h}_t} \sum_{\tau=t-l}^{\tau=t-1} \left( \prod_{\tau'=\tau}^{t-1} \frac{\partial \mathbf{h}_{\tau'+1}}{\partial \mathbf{h}_{\tau'}} \right) \delta_{\tau\ 0} = \frac{\partial L_t}{\partial \mathbf{h}_t} \mathbf{T}_t(\mathbf{h}_0) \quad (18)$$

We note that the sum conveniently drops as only the longest 'loop', in other words, the only summand that contributes is the product of Jacobians going from 0 to $t$. By considering this gradient, we can therefore ensure that no undesired signals stemming from shorter loops interfere with the analysis. Moreover, we note that we use the binary cross entropy loss which makes the derivative $\frac{\partial \mathcal{L}_t}{\partial \mathbf{h}_t}$ trivial.

In Fig 8 we show that *gradient flossing* boosts the gradient with respect to the initial conditions. Specifically, we compare two identical networks trained on the binary delayed temporal XOR task with a loop length of $d = 70$.

One network is trained with *gradient flossing* at epochs $e \in \{0, 100, 200, 300, 400\}$), while the other is trained without *gradient flossing*.

For the network without *gradient flossing*, the gradient norm of $|\frac{d\mathcal{L}}{dh_0}|$ diminishes to extremely small values ($< 10^{-6}$) and remains small throughout training. In contrast, for the network trained with *gradient flossing*, each episode of *gradient flossing* causes the norm $|\frac{d\mathcal{L}}{dh_0}|$ to spike, surpassing values larger than $10^{-2}$. These findings are direct evidence that *gradient flossing* boosts the gradient norm, facilitating to bridge long time horizons in challenging temporal credit assignment tasks. We observe that after several episodes of *gradient flossing*, the gradient $|\frac{d\mathcal{L}}{dh_0}|$ of the networks stays around $10^{-4}$ and eventually rise up to values around $10^{-2}$. Subsequent in training, the test accuracy surpasses chance level (Fig 8B). We observed this temporal relationship between gradient norm $|\frac{d\mathcal{L}}{dh_0}|$ and training success consistently across numerous network realizations (Fig 8C and D). These findings suggest that the gradient norm $|\frac{d\mathcal{L}}{dh_0}|$ can be a good predictor of learning success, sometimes hundreds of epochs before the accuracy exceeds the chance level of 50%. Indeed, when depicting the gradient norm aligned to the last epoch where accuracy was $\leq 50\%$, we see for many network realizations a gradual growth of gradient norm oven epochs before accuracy surpasses chance level (Fig 9A). Analogously, when plotting the accuracy as a function of epoch aligned with the last epoch with $|\frac{d\mathcal{L}}{dh_0}| < 0.001$, we observe for this task that the increase of gradient norm $|\frac{d\mathcal{L}}{dh_0}|$ reliably precedes the epoch at which the accuracy surpasses the chance level (See Fig 9B). We note that when measuring the overlap of the orientation of the gradient vector $\frac{d\mathcal{L}}{dh_0}$ with the first covariant Lyapunov vector of the forward dynamics, we found a significant increase in overlap around the training epoch where the accuracy surpasses the chance level both in networks with and without gradient flossing. This does not come as a surprise as the covariant Lyapunov vector measure the most unstable (or least stable) direction in the tangent space of a trajectory and perturbations of $h_0$ that have to travel over many epochs align

## D.1 *Gradient Flossing* Boosts Effective Dimension of Error Gradient

To further investigate the effect of *gradient flossing* on training, we investigated the structure of the error gradient and how it is changed by *gradient flossing*. To this end, we decompose the recurrent weight gradient $\sigma_i\left(\frac{d\mathcal{L}}{dW}\right)$ into in weighted sum of outer products using singular value decomposition (Fig 10).

As the Lyapunov exponents are the time-averaged logarithms of the singular values of the asymptotic long-term Jacobian $\mathbf{T}_t(\mathbf{h}_\tau)$, this allows us to directly link the effect of pushing Lyapunov exponents toward zero during *gradient flossing* to the structure of the error gradient of the recurrent weights, as they are intimately linked:

$$\frac{\partial \mathcal{L}_t}{\partial \mathbf{W}} = \frac{\partial \mathcal{L}_t}{\partial \mathbf{h}_t} \sum_{\tau=t-l}^{\tau=t-1} \left( \prod_{\tau'=\tau}^{t-1} \frac{\partial \mathbf{h}_{\tau'+1}}{\partial \mathbf{h}_{\tau'}} \right) \frac{\partial \mathbf{h}_\tau}{\partial \mathbf{W}} = \frac{\partial L_t}{\partial \mathbf{h}_t} \sum_\tau \mathbf{T}_t(\mathbf{h}_\tau) \frac{\partial \mathbf{h}_\tau}{\partial \mathbf{W}} \tag{19}$$

We again note that different 'loops' contribute to the total gradient expression and the Lyapunov exponents only characterize the longest loop. Further, we note that in our controlled tasks, depending on delay $d$, only few of the summands are relevant for solving the task. We thus expect the relevant gradient summand that carries important signals about the task to be contaminated by summands of both shorter and longer chains, which contribute irrelevant fluctuations.

The singular values of the recurrent weight gradient $\sigma_i\left(\frac{d\mathcal{L}}{dW}\right)$ as a function of training epoch reveal that the subordinate singular values subordinate singular value $\sigma_{20}$ and $\sigma_{40}$ exhibit peaks at the times of gradient flossing, while the first singular value $\sigma_1$ only shows a slight peak (Fig 10A). This indicates that *gradient flossing* increases the effective rank of the recurrent weight gradient $\frac{d\mathcal{L}}{dW}$. In other words, *gradient flossing* facilitates high-dimensional parameter updates. Our interpretation is, as *gradient flossing* pushes Lyapunov exponents to zero, the different summands in the total gradient contribute more equitable as long loops have neither a dominant contribution (which would happen for exploding gradients) nor a vanishing contribution (which would happen for vanishing gradients). This way, the sum of gradient terms has a higher effective rank.

In contrast, without *gradient flossing*, the subordinate singular values (in Fig 10A $\sigma_{20}$ and $\sigma_{40}$) rapidly diminish to extremely small values over training epochs and remain very small throughout training. Note however that the leading singular values $\sigma_1$ are of comparable size irrespective whether *gradient flossing* was performed or not.

We note that similar to the gradient norm of the loss with respect to the initial condition $|\frac{d\mathcal{L}}{dh_0}|$, the subordinate singular values seem to predict when the test accuracy of networks with *gradient flossing* grows beyond chance level (Fig 10B). We confirmed this in multiple other network realizations and give here another example we the accuracy grows beyond chance only later during training (Fig 11).

## D.2 *Gradient Flossing* Throughout Training Can Be Detrimental

We find that *gradient flossing* continued throughout all training epochs can be detrimental for performance (Fig 12). We demonstrate this again in the binary delayed temporal XOR task. We compare three different

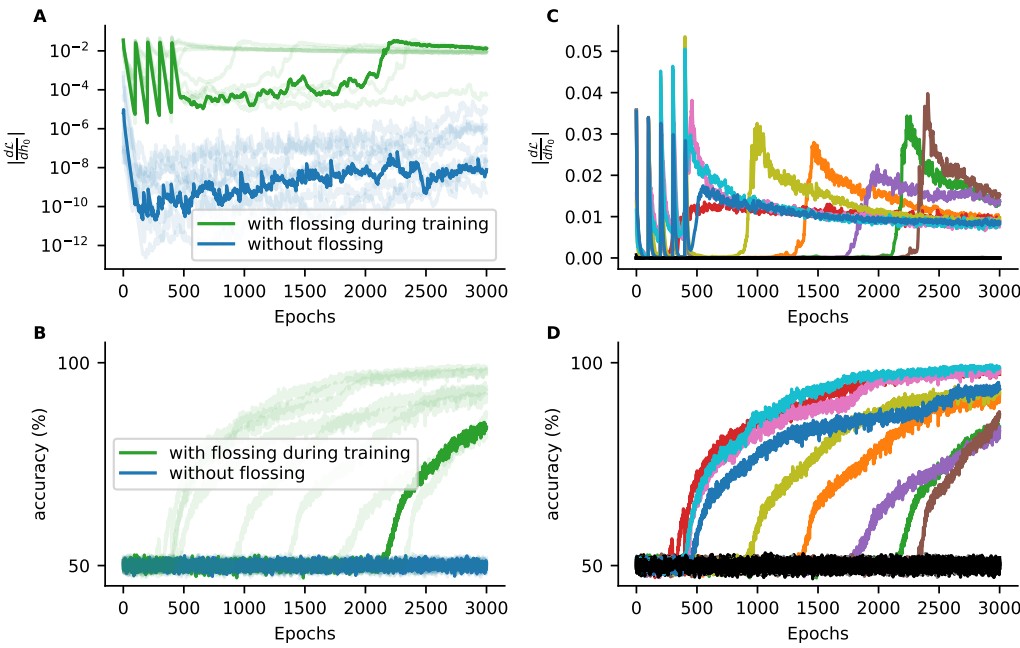

Figure 8: ***Gradient flossing* boosts norm of long-term Jacobian A)** Gradient norm of $\left|\frac{d\mathcal{L}}{dh_0}\right|$ as a function of training epochs for networks without *flossing* (blue) and networks with *flossing* during training (orange). Error gradient norm is boosted after *gradient flossing* at epochs $e \in \{0, 100, 200, 300, 400, 500\}$). In networks without *gradient flossing*, the gradient norms $\left|\frac{d\mathcal{L}}{dh_0}\right|$ are much smaller overall. One out of ten random network realizations with solid line, the other 9 with transparent line. **B)** Accuracy as a function of epoch, same depiction and network realizations as in **A**. Note that accuracy of networks with *gradient flossing* grows beyond chance level approximately when the gradient norm $\left|\frac{d\mathcal{L}}{dh_0}\right|$ becomes macroscopically large. **C)** Same as **A** in linear scale. Mean final test loss as a function of task difficulty (delay $d$) for delayed copy task. Different colors are different network realizations with *gradient flossing* during training. Black lines are without any *gradient flossing*. **D)** Accuracy as a function of epochs, same colors as in **C**. Note that for all network realizations the moments where gradient norm $\left|\frac{d\mathcal{L}}{dh_0}\right|$ becomes macroscopically large coincides with the moment the accuracy is beyond chance level. Parameters: $g = 1$, batch size $b = 16$, $N = 80$, epochs $= 10^4$, $T = 300$, *flossing* for $E_f = 500$ epochs on $k = 75$ Lyapunov exponents before training. Task: binary delayed XOR, delay $d = 70$, loss: cross entropy$(y, \hat{y})$.

conditions: Either, we *floss* throughout the training every 100 training epochs for 500 *flossing* epochs (red), or we *floss* only early during training at training epochs $e \in \{0, 100, 200, 300, 400\}$)(green) or we do not *floss* at all (blue).

We observe that after every episode of *gradient flossing*, the accuracy drops down close to chance level of 50% (Fig 12A red line). Between *flossing*, the accuracy quickly recovers but never reaches 100%. Simultaneously, when the accuracy drops the test error jumps up (Fig 12B). We also observed that the gradient norm $\left|\frac{d\mathcal{L}}{dh_0}\right|$ is initially boosted by *gradient flossing*, but stays close to indistinguishable once the gradient norm $\left|\frac{d\mathcal{L}}{dh_0}\right|$ becomes macroscopically large (Fig 12C). This suggests that once *gradient flossing* facilitates signal propagation across long time horizons and the network picks up the relevant gradient signal, further *gradient flossing* can be harmful to the actual task execution. We hypothesize that there might be (at least for the Vanilla networks considered here), a trade-off between the ability to bridge long time scales which seems to require one or several Lyapunov exponents of the forward dynamics close to zero and nonlinear tasks requirements, which require at least a fraction of the units to be in the nonlinear regime of the nonlinearity $\phi$, where $\phi'(x) < 1$. It would be an interesting future research avenue to further investigate this potential trade-off also in other network architectures.

### D.3 Lyapunov Exponents after Training With and Without *Gradient Flossing*

In Fig 13, we show the first (Fig 13A, B) and the tenth (Fig 13C, D) Lyapunov exponent after training on the spatial delayed XOR task both with and without *gradient flossing*. We find for successful networks with

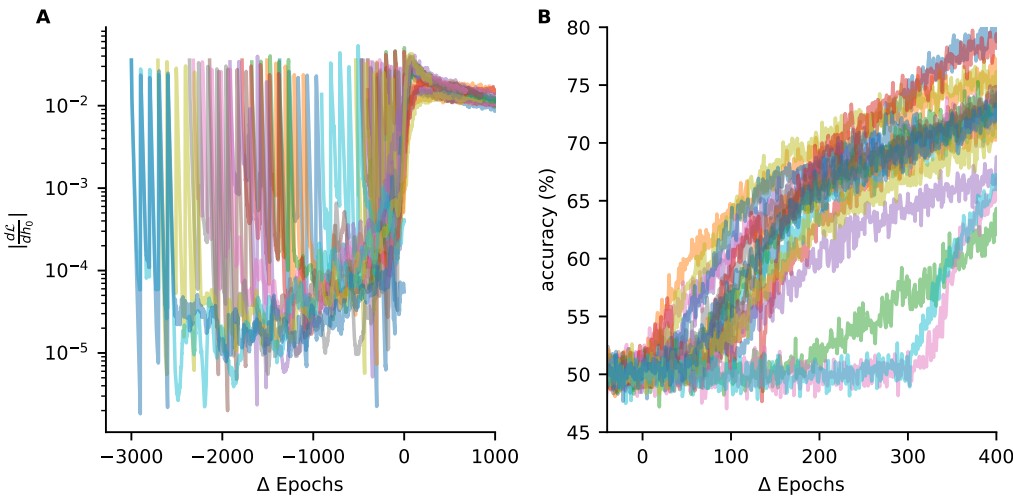

Figure 9: **Increase of gradient norm precedes epoch when accuracy exceeds chance level**
**A)** Gradient norm of $|\frac{d\mathcal{L}}{dh_0}|$ as a function of training epochs for 20 network realizations with *flossing* during training. Epochs are aligned to last epoch where accuracy is $\leq 50\%$. **B)** Same task and simulations as in **A**, but here accuracy as a function of epoch, for 20 network realizations with *flossing* during training. Epochs are aligned to the last epoch with $|\frac{d\mathcal{L}}{dh_0}| < 0.001$. Different colors are different network realizations. Parameters: $g = 1$, batch size $b = 16$, $N = 80$, epochs $= 10^4$, $T = 300$, *flossing* for $E_f = 500$ epochs on $k = 75$ Lyapunov exponents early during training at training epochs $e \in \{0, 100, 200, 300, 400\}$. Task: binary delayed XOR, delay $d = 70$, loss: cross entropy$(y, \hat{y})$.

*gradient flossing* a systematic relationship between the first Lyapunov exponent and the delay, that can be fitted by approximately by $\lambda_1(d) = -0.2exp.(-0.03delay)$. Unsuccessful networks with accuracy at chance level have a much smaller largest Lyapunov exponent. The same seems to hold true for the tenth Lyapunov exponent. In a previous study [63], a similar trend was observed, albeit in the context of a task that did not possess an analytically tractable temporal correlation structure, which might partially explain the less conclusive results. It is important to note that the numerical evaluation of Lyapunov exponents in recurrent LSTM networks in [63] was based solely on the $N \times N$ Jacobian of the memory state. From a dynamical systems standpoint, a $2N \times 2N$ Jacobian matrix encompassing interactions between both memory and cell states into account is required [9, 12, 64].

# E  *Gradient Flossing* for Linear Network

We provide code for *gradient flossing* in linear networks here. We find that *gradient flossing* also helps to train linear networks on tasks with many time steps that can be solved by linear networks, for example the copy task, but not for tasks the require a nonlinear input-output operation like the temporally delayed XOR task. Full analytical description of *gradient flossing* for linear networks would be a promising avenue for future research as networks with linear dynamics can still have nonlinear learning dynamics [23]. However this is beyond the cope of the presented work.

# F  Computational Complexity of Gradient Flossing

We present here a more in-depth scaling analysis of the computational cost of *gradient flossing*. There are three main contributors to the computational cost (table 1): First the RNN step, which has a computational complexity of $\mathcal{O}\left(N^2 b\right)$ per time step, where $N$ is the dimension of the recurrent network state (which in case of Vanilla networks equals the number of units) and $b$ is the batch-size both in the forward and backward pass. Second, the Jacobian step which scales with $\mathcal{O}\left(N^2 k\right)$ per time step, where $k$ is the number of *flossed* Lyapunov exponents. Third, the QR decomposition, which scales with $\mathcal{O}\left(N k^2\right)$, where $k$ is the number of Lyapunov exponents considered.

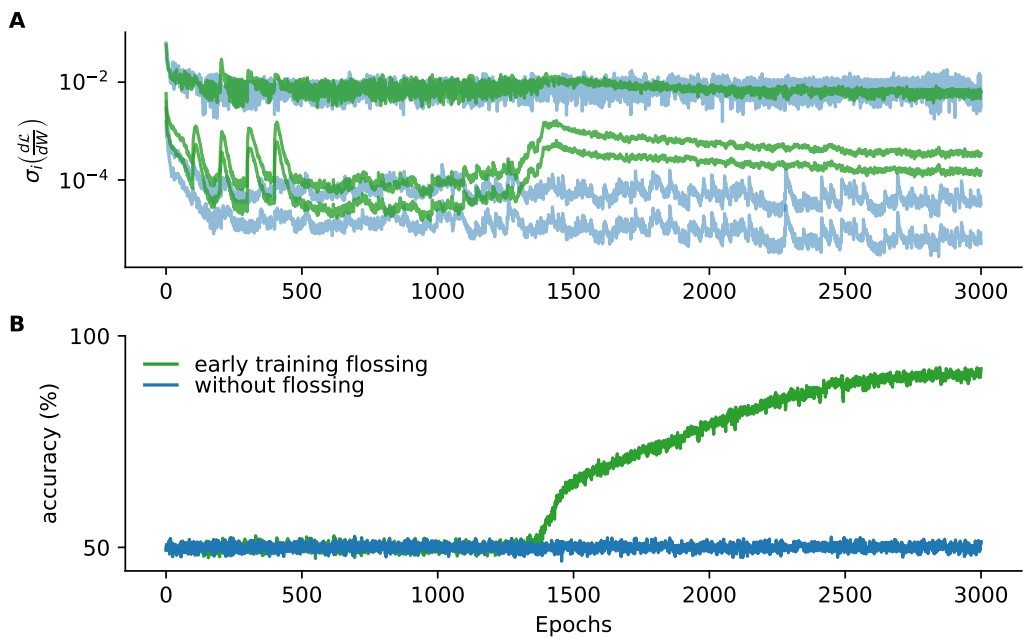

Figure 10: ***Gradient flossing* decreases condition number of recurrent weight error gradient A)** Singular values of recurrent weight gradient $\sigma_i \left( \frac{d\mathcal{L}}{dW} \right)$ as a function of training epochs for singular values $i \in 1, 20, 40$ for networks without *gradient flossing* (blue) and early training *gradient flossing* (green). At epochs of *gradient flossing*, the subordinate singular value $\sigma_{20}$ and $\sigma_{40}$ are peaked, while the first singular value $\sigma_1$ has only a slight peak. This indicates that *gradient flossing* increases the effective rank of the recurrent weight gradient $\frac{d\mathcal{L}}{dW}$. **B)** Accuracy as a function of training epochs. Note that accuracy of networks with *gradient flossing* grows beyond chance level approximately when the subordinate singular values singular value $\sigma_{20}$ and $\sigma_{40}$ are peaked increase, which enables high-dimensional parameters updates. Parameters: $g = 1$, batch size $b = 16$, $N = 80$, epochs $= 10^3$, $T = 300$, *gradient flossing* for $E_f = 500$ epochs on $k = 75$ Lyapunov exponents before training. Task: binary delayed XOR, delay $d = 70$, loss: cross entropy$(y, \hat{y})$.

Together, this results in a total amortized cost of $\mathcal{O}\left(N^2 \, b \, T\right)$ per training epoch, where $T$ is the number of training time steps and a total amortized costs per *flossing* epoch of $\mathcal{O}\left(N^2 \, T_f (1 + k/t_{\mathrm{ONS}} + k)\right)$ where $T_f$ is the number of *flossing* time steps.

In case of *preflossing*, thus, the total computation cost scale with $\mathcal{O}\left(N^2[Eb\,T + E_p \, T_f(1 + k/t_{\mathrm{ONS}} + k)]\right)$, where $E$ is the number of training epochs and $E_p$ is the number of *preflossing* epochs.

For *gradient flossing* during training (assuming that there is also *preflossing* done), the amortized cost scale with $\mathcal{O}\left(N^2[Eb\,T + E_p \, T_p + E_f \, T_f(1 + k/t_{\mathrm{ONS}} + k)]\right)$, where $E_f$ is the total number of *flossing* epochs during training.

Empirically, we find that both the number of *preflossing* epochs $E_p$ and *flossing* episodes $E_f$ necessary for training success is much smaller than the total number of training epochs $E$. For example, the *preflossing* for 500 epochs in the numerical experiment of Fig 3 took $\sim 37$ seconds, while the overall training on 10000 training epochs with batch size $b = 16$ took $\sim 1680$ seconds. Thus only approximately 2.2% of the total training time was spent on *gradient flossing*. Moreover, $T_p$ can be smaller than $T$, it just has to be long enough such that the temporal correlations in the task can be bridged. In case of the tasks discussed in the manuscript, this would be the delay $d$. It remains an important challenge to infer the suitable number of *flossing* time steps $T_f$ for tasks with unknown temporal correlation structure.

It would also be interesting to investigate how the CPU hours/wall-clock time/flops/Joule/$CO_2$-emission spent on *gradient flossing* vs on training networks with larger $N$ are trading off against each other. For this, we would suggest to first find the smallest network that on median successfully trains on a binary temporal XOR task for a fixed given delay $d$ and measure the computational resources involved in training it, e.g. in terms of CPU hours. Then compare it to a network with *gradient flossing*. This would be a promising analysis but is beyond our current computational budget. We will start such experiments an might be able to provide results during the reviewer period.

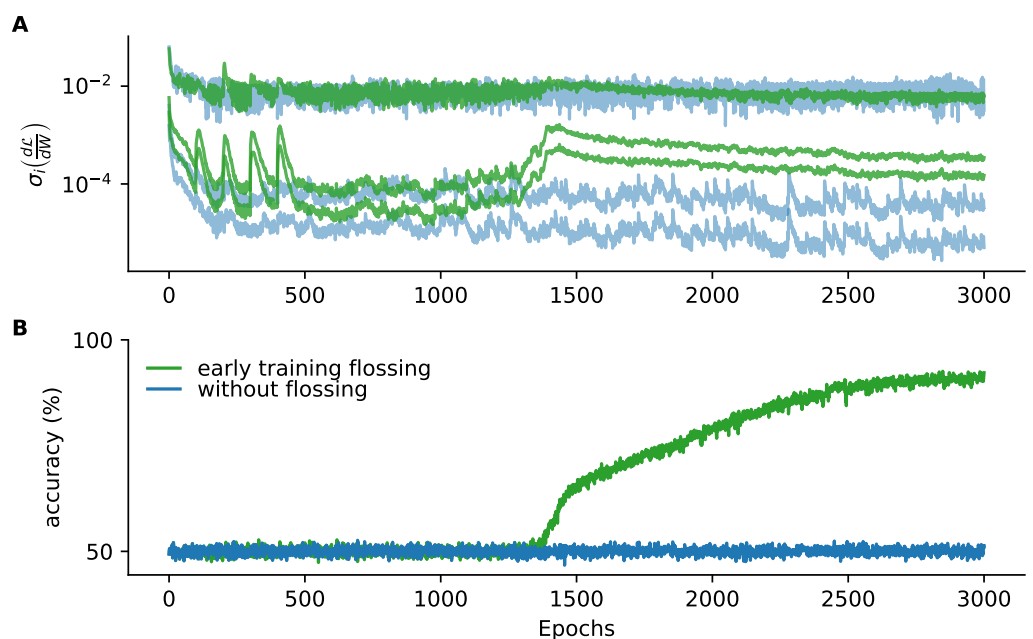

Figure 11: **Gradient flossing** **decreases condition number of recurrent weight error gradient** Same as Fig 10 for different network realization.

## G  Additional controls

We also investigate the effects of *gradient flossing* during the training with orthogonal weight initializations and confirm our finding that *gradient flossing* improves trainability on tasks that have long time horizons to bridge. Moreover, we find that *gradient flossing* during training can further improve trainability. We replicated the two more challenging tasks from the main paper (Fig 4) for orthogonal initialization with variable temporal complexity and performed *gradient flossing* either both during and before training, only before training, or not at all.

Fig 14A shows the test accuracy for Vanilla RNNs with orthogonal initialization trained on the delayed temporal XOR task $y_t = x_{t-d/2} \oplus x_{t-d}$ with random Bernoulli process $x \in \{0, 1\}$. The accuracy of orthogonal Vanilla RNNs falls to chance level for $d \geq 40$ (Fig 14C). With *gradient flossing* before training, the trainability can be improved, but still falls close to chance level for $d = 70$. In contrast, for initially orthogonal networks with *gradient flossing* during training, the accuracy is improved to $> 80\%$ at $d = 70$. In this case, we *preflossed* for 500 epochs before task training and again after 500 epochs of training on the task. In Fig 14B, D the networks have to perform the nonlinear XOR operation $y_t = x_{t-d}^1 \oplus x_{t-d}^2 \oplus x_{t-d}^3$ on a three-dimensional binary input signal $x^1$, $x^2$, and $x^3$ and generate the correct output with a delay of $d$ steps identical to Fig 4 in the main text. Again, we observe similar to networks with Gaussian initialization that *flossing* before training improves the performance compared to baseline, but starts failing for long delays $d > 60$. In contrast, orthogonal networks that are also *flossed* during training can solve even more difficult tasks (Fig 14D). We note that for Fig 14B and D, we trained the network only on 5000 epochs, compared to 10000 epochs in networks with random Gaussian initialization because for 10000 epochs, both networks with *gradient flossing* only before training and with *gradient flossing* before and during training were able to bridge $d = 70$. These results suggest that orthogonal initialization does seem to slightly improve performance for tasks with long time horizons to bridge and *gradient flossing* and additionally boost the performance. Thus orthogonal initialization and *gradient flossing* seems to go well together. It would be interesting to study if orthogonal initialization also reduces the number of *gradient flossing* steps necessary to improve performance.

## H  Additional Details on Training Tasks

In this section, we provide a more rigorous definition of the tasks used for training, as discussed in Section 3:

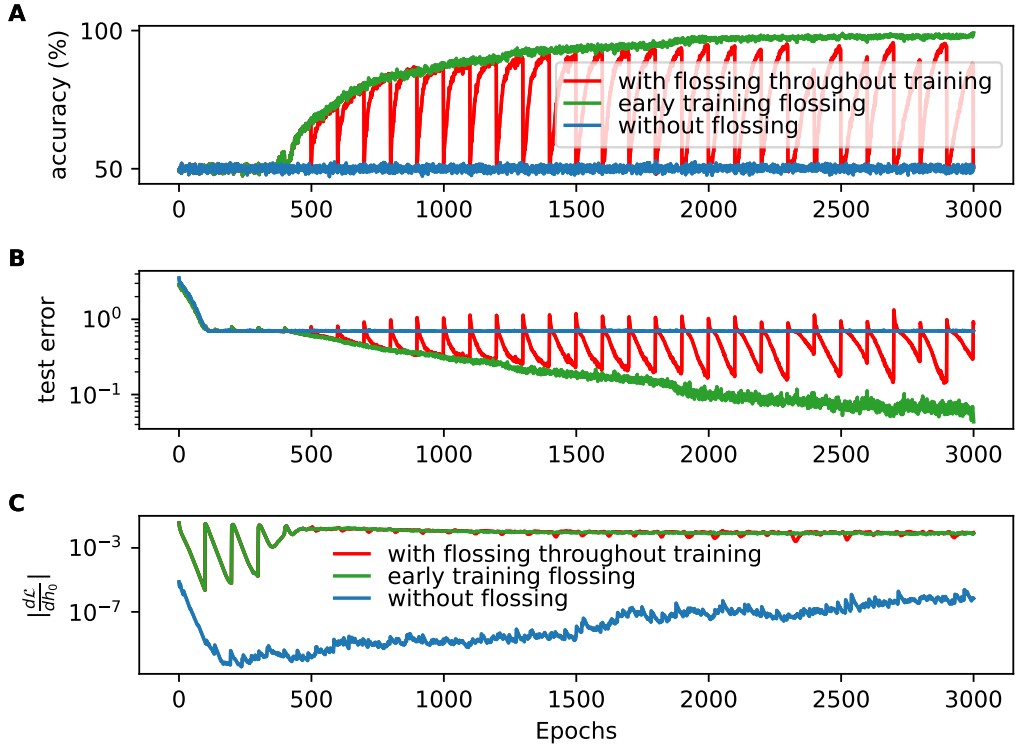

Figure 12: ***Gradient flossing* throughout training can be detrimental to learning A)** Accuracy as a function of training epochs for binary temporal delayed XOR task for *gradient flossing* throughout training every 100 training epochs (red). Accuracy drops down close to chance level every time after *gradient flossing* but recovers quickly between. Same for only 5 episodes of *gradient flossing* at epochs $e \in \{0, 100, 200, 300, 400\}$) (green) and no *flossing* at all (blue). **B)** Test error as a function of training epochs. **C)** Gradient norm of $\left|\frac{d\mathcal{L}}{dh_0}\right|$ as a function of training epochs for networks without *gradient flossing* (blue) and networks with *flossing* throughout training (red) and early training *gradient flossing* (green). Error gradient norm is boosted after each *gradient flossing*. In networks without *gradient flossing*, the gradient norms $\left|\frac{d\mathcal{L}}{dh_0}\right|$ are much smaller overall. Parameters: $g = 1$, batch size $b = 16$, $N = 80$, epochs $= 10^4$, $T = 300$, *gradient flossing* for $E_f = 500$ epochs on $k = 75$ Lyapunov exponents before training. Task: binary delayed XOR, delay $d = 70$, loss: cross entropy$(y, \hat{y})$.

## H.1 Copy task

For the copy task, the target network readout at time $t$ is $y_t = x_{t-d}$, where $d$ denotes the delay. We chose the input to be sampled i.i.d. from a uniform distribution between 0 and 1.

## H.2 Temporal XOR task

The temporal XOR task requires the target network readout $y_t$ at time $t$ to be computed as follows:

$$y_t = |x_{t-d/2} - x_{t-d}| \tag{20}$$

where again $d$ denotes a time delay of $d$ time steps. In the case of $x \in \{0, 1\}^2$ and $y \in \{0, 1\}$, the output $y_t$ follows the truth table of the XOR digital logic gate (Table 2). Thus, the function $f(x_a, x_b) = |x_a - x_b|$ can be seen as an analytical representation of the XOR gate. It is important to note that $f(x, 0) = x$ only for $x \geq 0$, and that this task requires a nonlinearity. The implementation can easily be constructed analytically, for example, using two rectified linear units $\phi(x) = \max(x, 0)$ the outbut can be constructed by

$$f(x_a, x_b) = |x_a - x_b| = \phi(x_a - x_b) + \phi(x_b - x_a). \tag{21}$$

Together with a delay line to transmit the signal $x_{t-d}$ over time, this can solve the task.

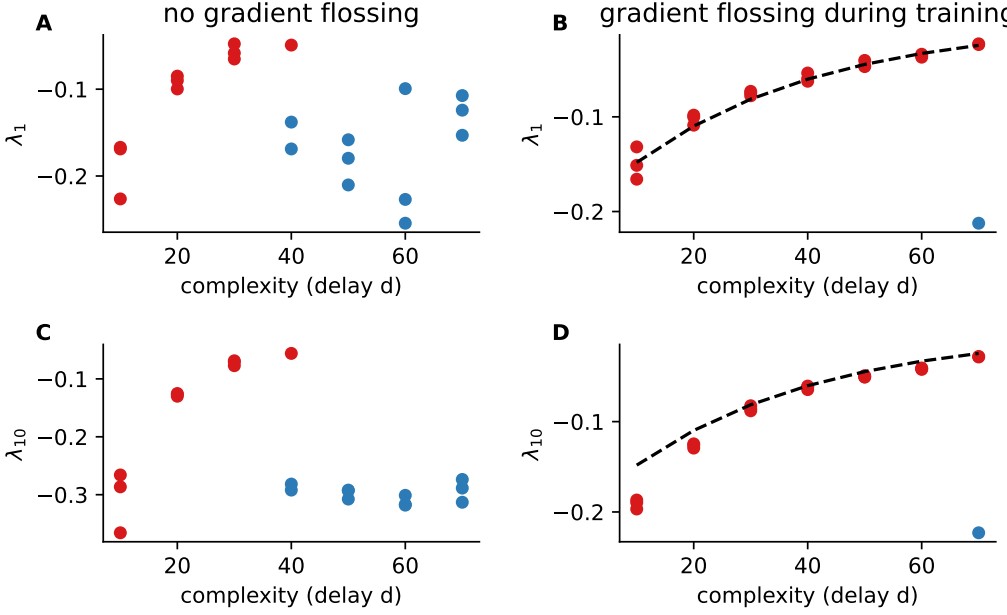

Figure 13: **Lyapunov exponents of trained networks with and without *gradient flossing* A)** First Lyapunov exponents $\lambda_1$ for Vanilla networks trained on spatial delayed XOR task as a function of the delay with no *gradient flossing*. Colored-coded is test accuracy at the end of training where red corresponds to 100% accuracy and blue to chance level (50%). **B)** Same as **A** for networks with *gradient flossing* during training. Black dashed line shows that Lyapunov exponents of successfully trained networks can be approximated by the empirical fit $\lambda_1(d) = -0.2 exp.(-0.03 delay)$. (Protocol for *gradient flossing* during training same as main text Fig 4B). **C)** Same as **A** for tenth Lyapunov exponents $\lambda_{10}$. **D)** Same as **B** for tenth Lyapunov exponents $\lambda_{10}$. Same fit as in **B** also describes $\lambda_{10}$. Parameters: $g = 1$, batch size $b = 16$, $N = 80$, epochs $= 10^4$, $T = 300$, *gradient flossing* for $E_f = 500$ epochs on $k = 75$ Lyapunov exponents before training. Task: binary spatial delayed XOR, loss: cross entropy$(y, \hat{y})$.

# I   Additional Background on Lyapunov Exponents of RNNs

An autonomous dynamical system is usually defined by a set of ordinary differential equations $d\mathbf{h}/dt = \mathbf{F}(\mathbf{h})$, $\mathbf{h} \in \mathbb{R}^N$ in the case of continuous-time dynamics, or as a map $\mathbf{h}_{s+1} = \mathbf{f}(\mathbf{h}_s)$ in the case of discrete-time dynamics. In the following, the theory is presented for discrete-time dynamical systems for ease of notation, but everything directly extends to continuous-time systems [49]. Together with an initial condition $\mathbf{h}_0$, the map forms a trajectory. As a natural extension of linear stability analysis, one can ask how an infinitesimal perturbation $\mathbf{h}_0' = \mathbf{h}_0 + \epsilon \mathbf{u}_0$ evolves in time. Chaotic systems are sensitive to initial conditions; almost all infinitesimal perturbations $\epsilon \mathbf{u}_0$ of the initial condition grow exponentially with time $|\epsilon \mathbf{u}_t| \approx \exp(\lambda_1 t)|\epsilon \mathbf{u}_0|$. Finite-size perturbations, therefore, may lead to a drastically different subsequent behavior. The largest Lyapunov exponent $\lambda_1$ measures the average rate of exponential divergence or convergence of nearby initial conditions:

$$\lambda_1(\mathbf{h}_0) = \lim_{t \to \infty} \frac{1}{t} \lim_{\epsilon \to 0} \log \frac{||\epsilon \mathbf{u}_t||}{||\epsilon \mathbf{u}_0||} \tag{22}$$

In dynamical systems that are ergodic on the attractor, the Lyapunov exponents do not depend on the initial conditions as long as the initial conditions are in the basins of attraction of the attractor. Note that it is crucial to first take the limit $\epsilon \to 0$ and then $t \to \infty$, as $\lambda_1(\mathbf{h}_0)$ would be trivially zero for a bounded attractor if the limits are exchanged, as $\lim_{t \to \infty} \log \frac{||\epsilon \mathbf{u}_t||}{||\epsilon \mathbf{u}_0||}$ is bounded for finite perturbations even if the system is chaotic. To measure $k$ Lyapunov exponents, one has to study the evolution of $k$ independent infinitesimal perturbations $\mathbf{u}_s$ spanning the tangent space:

$$\mathbf{u}_{s+1} = \mathbf{D}_s \mathbf{u}_s \tag{23}$$

|  | forward pass | backward pass |
|---|---|---|
| RNN dynamics | $\mathcal{O}\left(N^2\, b\right)$ | " |
| Jacobian step | $\mathcal{O}\left(N^2\, k\right)$ | " |
| QR step | $\mathcal{O}\left(N\, k^2\right)$ | " |
| total amortized costs per training epoch | $\mathcal{O}\left(N^2\, b\, T\right)$ | " |
| total amortized costs per *gradient flossing* epoch | $\mathcal{O}\left(N^2\, T_f(1 + k/t_{\mathrm{ONS}} + k)\right)$ | " |
| **total amortized costs of *preflossing*** | $\mathcal{O}\left(N^2[Eb\, T + E_p\, T_f(1 + k/t_{\mathrm{ONS}} + k)]\right)$ | " |
| **total amortized costs *flossing* during training** | $\mathcal{O}\left(N^2[Eb\, T + E_p\, T_p + E_f\, T_f(1 + k/t_{\mathrm{ONS}} + k)]\right)$ | " |

Table 1: **Computational cost for *gradient flossing* and training of RNNs**
$N$ denotes number of neurons, $b$ is the batch size, $T$ is the number of time steps in forward pass of training, $T_f$ is the number of time steps in forward pass of *flossing*, $t_{\mathrm{ONS}}$ is the reorthonormalization interval, $k$ is the number of *flossed* Lyapunov exponents, $E$ is the number of training epochs, $E_p$ is the number of *preflossing* epochs, $E_f$ is the number of *flossing* epochs during training. Empirically, we find that the necessary number of *preflossing* epochs $E_p$ and *flossing* episodes $E_f$ is much smaller than both the total number of training epochs $E$. Moreover, $T_p$ can be smaller than $T$.

Table 2: XOR

| input $x_{t-d}$ | input $x_{t-2d}$ | target output $y_t$ |
|---|---|---|
| 0 | 0 | 0 |
| 0 | 1 | 1 |
| 1 | 0 | 1 |
| 1 | 1 | 0 |

where the $N \times N$ Jacobian $\mathbf{D}_s(\mathbf{h_s}) = \mathrm{d}\mathbf{f}(\mathbf{h_s})/\mathrm{d}\mathbf{h}$ characterizes the evolution of generic infinitesimal perturbations during one step. Note that this Jacobian along the trajectory is equivalent to a stability matrix only at a fixed point, i.e., when $\mathbf{h}_{s+1} = \mathbf{f}(\mathbf{h}_s) = \mathbf{h}_s$.

We are interested in the asymptotic behavior, and therefore we study the long-term Jacobian

$$\mathbf{T}_t(\mathbf{h}_0) = \mathbf{D}_{t-1}(\mathbf{h}_{t-1})\ldots\mathbf{D}_1(\mathbf{h}_1)\mathbf{D}_0(\mathbf{h}_0). \tag{24}$$

Note that $\mathbf{T}_t(\mathbf{h}_0)$ is a product of generally noncommuting matrices. The Lyapunov exponents $\lambda_1 \geq \lambda_2 \cdots \geq \lambda_N$ are defined as the logarithms of the eigenvalues of the Oseledets matrix

$$\mathbf{\Lambda}(\mathbf{h}_0) = \lim_{t \to \infty}[\mathbf{T}_t(\mathbf{h}_0)^\top \mathbf{T}_t(\mathbf{h}_0)]^{\frac{1}{2t}}, \tag{25}$$

where $\top$ denotes the transpose operation. The expression inside the brackets is the Gram matrix of the long-term Jacobian $\mathbf{T}_t(\mathbf{h}_0)$. Geometrically, the determinant of the Gram matrix is the squared volume of the parallelotope spanned by the columns of $\mathbf{T_t(h_0)}$. Thus, the exponential volume growth rate is given by the sum of the logarithms of its first $k$ (sorted) eigenvalues. Oseledets' multiplicative ergodic theorem guarantees the existence of the Oseledets matrix $\mathbf{\Lambda}(\mathbf{h}_0)$ for almost all initial conditions $\mathbf{h}_0$ [48]. In ergodic systems, the Lyapunov exponents $\lambda_i$ do not depend on the initial condition $\mathbf{h}_0$. However, for a numerical calculation of the Lyapunov spectrum, Eq 25 cannot be used directly because the long-term Jacobian $T_t(\mathbf{h}_0)$ quickly becomes ill-conditioned, i.e., the ratio between its largest and smallest singular value diverges exponentially with time.

## J    Algorithm for Calculating Lyapunov Spectrum of Rate Networks

For calculating the first $k$ Lyapunov exponents, we exploit the fact that the growth rate of a $k$-dimensional infinitesimal volume element is given by $\lambda^{(m)} = \sum_{i=1}^{m} \lambda_i$. Therefore, $\lambda_1 = \lambda^{(1)}$, $\lambda_2 = \lambda^{(2)} - \lambda_1$, $\lambda_3 = \lambda^{(3)} - \lambda_1 - \lambda_2$, ... [50]. The volume growth rates can be obtained via QR-decomposition.

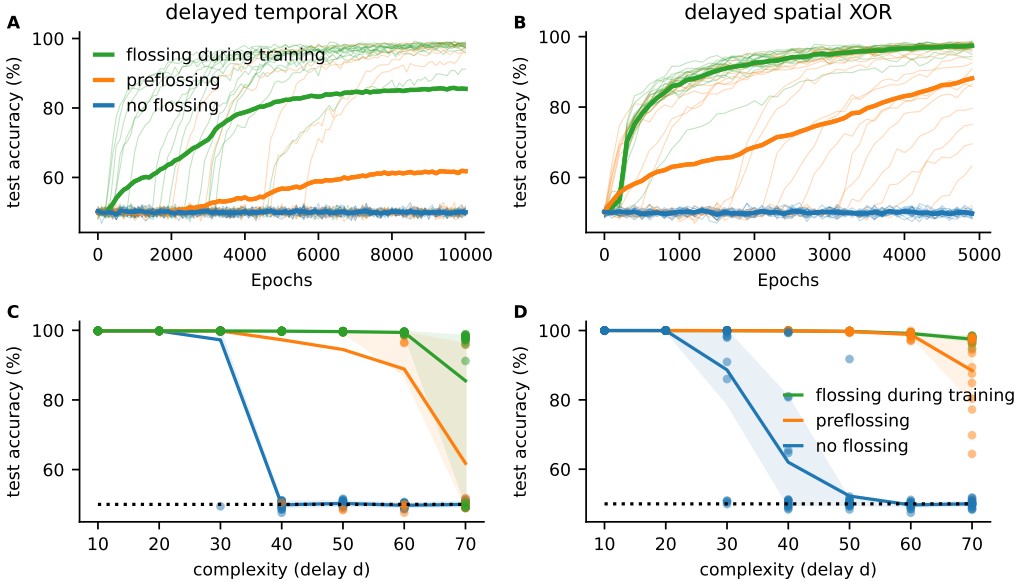

Figure 14: ***Gradient flossing* before and during training improves trainability for orthogonal nets**
**A)** Test accuracy for orthogonally initialized vanilla RNNs trained on delayed temporal binary XOR task $y_t = x_{t-d/2} \oplus x_{t-d}$ with *gradient flossing* during training (green), *preflossing* (orange), and with no *gradient flossing* (blue) for $d = 70$. Solid lines are mean, transparent thin lines are individual network realizations **B)** Same as **A** for delayed spatial XOR task with $y_t = x_{t-d}^1 \oplus x_{t-d}^2 \oplus x_{t-d}^3$ . **C)** Test accuracy as a function of task difficulty (delay $d$) for delayed temporal XOR task. **D)** Test accuracy as a function of task difficulty (delay $d$) for delayed spatial XOR task. Parameters: $g = 1$, batch size $b = 16$, $N = 80$, epochs $= 10^4$ for delayed temporal XOR, epochs $= 5000$ for delayed spatial XOR, $T = 300$, *flossing* for $E_f = 500$ epochs on $k = 75$ Lyapunov exponents before training and during training for green lines, and only before training for orange lines. Shaded areas are 25% and 75% percentiles, solid lines are means, transparent dots are individual simulations, task loss is cross-entropy between $y, \hat{y}$.

First, we evolve an initially orthonormal system $\mathbf{Q}_s = [\mathbf{q}_s^1, \mathbf{q}_s^2, \dots \mathbf{q}_s^m]$ in the tangent space along the trajectory using the Jacobian $\mathbf{D}_s$:

$$\widetilde{\mathbf{Q}}_{s+1} = \mathbf{D}_s \mathbf{Q}_s \tag{26}$$

A continuous system can be transformed into a discrete system by considering a stroboscopic representation, where the trajectory is only considered at certain discrete time points. We use here the notation of discrete dynamical systems where this corresponds to performing the product of Jacobians along the trajectory $\widetilde{\mathbf{Q}}_{s+1} = \mathbf{D}_s \mathbf{Q}_s$. We study the discrete network dynamics in the limit of small time step $\Delta t \to 0$ and for discrete time $\Delta t = 1$. The notation can be readily extended to continuous systems [49].

Second, we extract the exponential growth rates using the QR-decomposition,

$$\widetilde{\mathbf{Q}}_{s+1} = \mathbf{Q}_{s+1} \mathbf{R}^{s+1},$$

which uniquely decomposes $\widetilde{\mathbf{Q}}_{s+1}$ into an orthonormal matrix $\mathbf{Q}_{s+1}$ of size $N \times k$ so $\mathbf{Q}_{s+1}^\top \mathbf{Q}_{s+1} = \mathbb{1}_{m \times m}$ and to an upper triangular matrix $\mathbf{R}^{s+1}$ of size $k \times k$ with positive diagonal elements. Geometrically, $\mathbf{Q}_{s+1}$ describes the rotation of $\mathbf{Q}_s$ caused by $\mathbf{D}_s$ and the diagonal entries of $\mathbf{R}^{s+1}$ describe the stretching and shrinking of the columns of $\mathbf{Q}_s$, while the off-diagonal elements represent the shearing. Fig 15 visualizes $\mathbf{D}_s$ and the QR-decomposition for $k = 2$.

The Lyapunov exponents are given by time-averaged logarithms of the diagonal elements of $\mathbf{R}^s$:

$$\lambda_i = \lim_{t \to \infty} \frac{1}{t} \log \prod_{s=1}^{t} \mathbf{R}_{ii}^s = \lim_{t \to \infty} \frac{1}{t} \sum_{s=1}^{t} \log \mathbf{R}_{ii}^s. \tag{27}$$

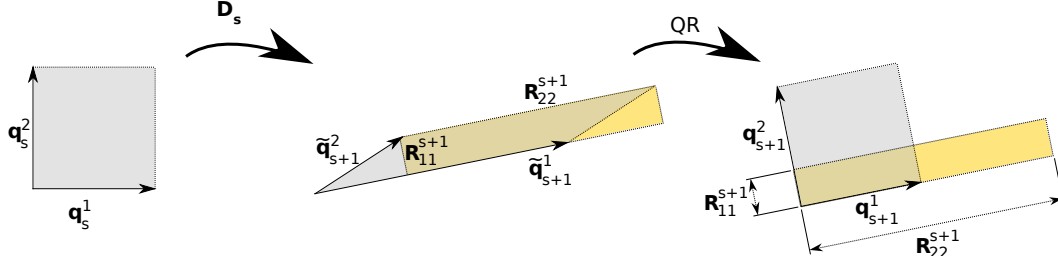

Figure 15: **Geometric illustration of Lyapunov spectrum calculation.** An orthonormal matrix $\mathbf{Q}_s = [\mathbf{q}_s^1, \mathbf{q}_s^2, \dots \mathbf{q}_s^m]$, whose columns are the axes of an $k$-dimensional cube, is rotated and distorted by the Jacobian $\mathbf{D}_s$ into an $k$-dimensional parallelotope $\widetilde{\mathbf{Q}}_{s+1} = \mathbf{D}_s \mathbf{Q}_s$ embedded in $\mathbb{R}^\mathbf{N}$. The figure illustrates this for $k = 2$, in which case the columns of $\widetilde{\mathbf{Q}}_{s+1}$ span a parallelogram, which can be divided into a right triangle and a trapezoid and rearranged into a rectangle. Thus, the area of the gray parallelogram is the same as that of the orange rectangle. The QR-decomposition reorthonormalizes $\widetilde{\mathbf{Q}}_{s+1}$ by decomposing it into the product of an orthonormal matrix $\mathbf{Q}_{s+1} = [\mathbf{q}_{s+1}^1, \mathbf{q}_{s+1}^2, \dots \mathbf{q}_{s+1}^m]$ and the upper-triangular matrix $\mathbf{R}^{s+1}$. $\mathbf{Q}_{s+1}$ describes the rotation of $\mathbf{Q}_s$ caused by $\mathbf{D}_s$. The diagonal entries of $\mathbf{R}^{s+1}$ gives the stretching/shrinking along the columns of $\mathbf{Q}_{s+1}$, thus the volume of the parallelotope formed by the first $k$ columns of $\widetilde{\mathbf{Q}}_{s+1}$ is given by $V_m = \prod_{i=1}^m \mathbf{R}_{ii}^{s+1}$. The time-averaged logarithms of the diagonal elements of $\mathbf{R}^s$ give the Lyapunov spectrum: $\lambda_i = \lim_{t_\text{sim} \to \infty} \frac{1}{t_\text{sim}} \log \prod_{s=1}^t \mathbf{R}_{ii}^s = \lim_{t_\text{sim} \to \infty} \frac{1}{t} \sum_{s=1}^t \log \mathbf{R}_{ii}^s$.

Note that the QR-decomposition does not need to be performed at every simulation step, just sufficiently often, i.e., once every $s_\text{ONS}$ steps such that $\widetilde{\mathbf{Q}}_{s+s_\text{ONS}} = \mathbf{D}_{s+s_\text{ONS}-1} \cdot \mathbf{D}_{s+s_\text{ONS}-2} \dots \mathbf{D}_s \cdot \mathbf{Q}_s$ remains well-conditioned [50]. An appropriate reorthonormalization interval $s_\text{ONS} = t_\text{ONS}/\Delta t$ thus depends on the condition number, the ratio of the smallest and largest singular value:

$$\kappa_2(\widetilde{\mathbf{Q}}_{s+s_\text{ONS}}) = \kappa_2(\mathbf{R}^{s+s_\text{ONS}}) = \frac{\sigma_1(\mathbf{R}^{s+s_\text{ONS}})}{\sigma_m(\mathbf{R}^{s+s_\text{ONS}})} = \frac{\mathbf{R}_{11}^{s+s_\text{ONS}}}{\mathbf{R}_{mm}^{s+s_\text{ONS}}}. \tag{28}$$

An initial transient should be disregarded in the calculation of the Lyapunov spectrum because $\mathbf{h}$ first has to converge towards the attractor and $\mathbf{Q}$ has to converge to the unique eigenvectors of the Oseledets matrix (Eq 25) [60]. A simple example of this algorithm in pseudocode is:

---

**Algorithm 2 Jacobian-based algorithm for Lyapunov spectrum**

---

    initialize $\mathbf{h}$, $\mathbf{Q}$
    evolve $\mathbf{h}$ until it is on attractor (avoid initial transient)
    evolve $\mathbf{Q}$ until it converges to the eigenvectors of the backward Oseledets matrix
    set $\gamma_i = 0$
    **for** $t = 1 \to T$ **do**
        $\mathbf{h} \leftarrow \mathbf{f}(\mathbf{h})$
        $\mathbf{D} \leftarrow \frac{d\mathbf{f}}{d\mathbf{h}}$
        $\mathbf{Q} \leftarrow \mathbf{D} \cdot \mathbf{Q}$
        **if** $s \equiv 0 \pmod{s_\text{ONS}}$ **then**
            $\mathbf{Q}, \mathbf{R} \leftarrow \text{qr}(\mathbf{Q})$
            $\gamma_i \mathrel{+}= \log(R_{ii})$
        **end if**
    **end for**
    $\lambda_i = \gamma_i/T$

---

It is guaranteed that under general conditions initially random orthonormal systems will exponentially converge towards a unique basis that is given by the eigenvectors of the Oseledets matrix Eq 25 [60]. A minimal example of this algorithm in pseudocode is shown in appendix 3. A feasible strategy to determine the reorthonormalization time interval $t_\text{ONS}$ is to get first a rough estimate of the Lyapunov spectrum using a short simulation time $t_\text{sim}$ and a small $t_\text{ONS}$ and repeat with a longer simulation time and a $t_\text{ONS}$ based on the Lyapunov spectrum of the rough estimate of the Lyapunov spectrum. Another strategy is, to first iteratively adapt $t_\text{ONS}$ on a short simulation run to get an acceptable condition number. It should be noted that there exists a diversity of other methods to estimate the Lyapunov spectrum [14, 49, 79, 80].

# K  Convergence of Lyapunov Exponents of RNNs

In Fig. 16, we demonstrate the convergence of the Lyapunov exponents. We show the estimate of the Lyapunov exponents $\lambda_i$ for $i = 1, 20, 60, 80$ for different initial conditions but identical network realization.

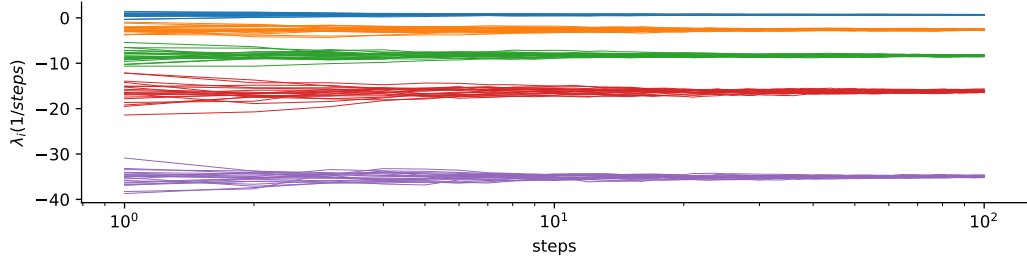

Figure 16: **Convergence of Lyapunov exponents** Convergence of selected Lyapunov exponents $\lambda_i$ for ten identical network realizations with different initial conditions with simulation time ($i = 1, 20, 60, 80$) for $\sigma = 1$ and $g = 1$. (Other parameters: $N = 80$, $t_{\mathrm{sim}} = 100$ steps, $t_{\mathrm{ONS}} = 1$).

