# OpenReview forum: "Gradient Flossing: Improving Gradient Descent through Dynamic Control of Jacobians"
_NeurIPS.cc/2023/Conference — NeurIPS 2023 poster_

### Official Review · Reviewer_5Z2M · 2023-07-04

**Soundness:** 3 good
**Presentation:** 3 good
**Contribution:** 2 fair
**Rating:** 4
**Confidence:** 4

**Summary:**

The authors propose a novel approach called "gradient flossing" to tackle the instability of gradients in the training of recurrent neural networks (RNNs). The process of pushing the Lyapunov exponents toward zero is referred to as "flossing" the gradients. This stabilizes the gradients and improves network training.

**Strengths:**

+ The authors show improvements in several toy sequence learning tasks: Delayed Copy Task, Temporal XOR Task, etc.

+ Idea of using Lyapunov exponents towards zero to mitigate vanishing and exploding gradients is well justified and works as expected on toy tasks.



**Weaknesses:**

The lack of any experiments on non-simulated data is problematic, and the explanation given by the authors is not convincing. The paper's authors explicitly state that they deliberately did not use sequential MNIST or similar non-toy tasks commonly used to probe exploding/vanishing gradients.

> We deliberately do not use sequential MNIST or similar toy tasks commonly used to probe exploding/vanishing gradients, because we want a task where the structure of long-range dependencies in the data is transparent and can be varied ad libitum

While toy tasks are more transparent, real-world data is important to understand the proposed methods' potential failure modes and expected characteristics in *uncontrolled* settings. Avoiding running experiments on the basis that the results are more difficult to interpret than simulated data is not an acceptable excuse. Transparency in the work itself is hindered when experiments are consciously avoided.

**Questions:**

Can the authors comment that such a technique of pushing the Lyapunov exponents to zero could limit the capacity to learn certain tasks? Maybe enabling the network to amplify or forget things is important for some task contexts, which has been a motivation for specialized architectures like the LSTM.

**Limitations:**

Experimental limitation mentioned above.

---

> ### Author Rebuttal · Authors · 2023-08-09
>
>
>
> Thank you for your thoughtful review and feedback on our manuscript. We appreciate the acknowledgment and recognition of our novel approach involving the use of Lyapunov exponents towards zero to address the challenges of vanishing and exploding gradients in RNNs.
>
> > While toy tasks are more transparent, real-world data is important to understand the proposed methods' potential failure modes and expected characteristics in uncontrolled settings. Avoiding running experiments on the basis that the results are more difficult to interpret than simulated data is not an acceptable excuse. Transparency in the work itself is hindered when experiments are consciously avoided.
>
> We recognize the importance of assessing our method on real-world datasets and thank the reviewer for emphasizing this aspect. Our rationale behind beginning with synthetic toy tasks was to "open the black box" and comprehensively understand the tangent space structure of gradient descent from a dynamical systems perspective and how it is improved by gradient flossing. This rigorous examination has allowed us to delve into not just vanishing and exploding gradients, but also:
>
> - Observe how the complexity of tasks impacts the number of required Lyapunov exponents to be flossed, as showcased in Supplementary figure 2.
>
> - Study how gradient flossing enhances the norm of the long-term Jacobian (Supplementary figure 7).
>
> - Establish a relationship between the training epoch, where accuracy exceeds the chance level, and the increase in the gradient norm (Supplementary figure 8).
>
> - Analyze the positive effects of gradient flossing on the condition number of the recurrent weight gradient (Figure 2 and supplementary figure 8).
>
> - Formulate a mathematical correlation between the post-gradient-flossed Lyapunov exponents and task complexity, quantified by the delay \(d\) (Supplementary figure 9).
>
> To add, our main contribution hinges on the connection between the condition number of the long-term Jacobian and the Lyapunov spectrum. Our work, especially in the section 5 on "the condition number of the Long-Term Jacobian" (Figure 2), provides novel insights into the dimensionality of the gradient update, offering an analytical perspective on the nature of signal propagation backward in time during RNN training.
>
> > Can the authors comment that such a technique of pushing the Lyapunov exponents to zero could limit the capacity to learn certain tasks? Maybe enabling the network to amplify or forget things is important for some task contexts, which has been a motivation for specialized architectures like the LSTM.
>
> You've raised an excellent point. Our analysis in Supplementary figure 11 demonstrates potential conflicts of gradient flossing with certain task goals, where persistent gradient flossing over multiple Lyapunov exponents may inhibit the RNN's nonlinearity. We've addressed this challenge by limiting gradient flossing steps and subsequently focusing on task loss minimization without gradient flossing. For architectures like LSTM, this scenario could differ, as their latent variables can effectively carry gradient information over extended time frames, even amidst active nonlinearities on \(h\).
>
> In conclusion, we deeply value the feedback and are keen to explore the applicability and nuances of our gradient flossing technique on real-world datasets beyond toy tasks, to make our contribution more comprehensive and robust in the field of RNN training.

---

> > ### Comment · Reviewer_5Z2M · 2023-08-15
> >
> > Having reviewed the rebuttals of the authors to my questions and weaknesses and also reviewing rebuttals to other reviewers, I will keep my score. There is still the importance of demonstrating this method on data that the overall research community is familiar with, and the authors do not seem willing to demonstrate this while still claiming to solve a well-known problem in the community of vanishing/exploding gradients. I still consider this a severe weakness of this work.
> >
> > The request is not to see this method as performant or superior to other work. It is to understand the proposed methods' potential failure modes and expected characteristics in uncontrolled settings.

---

> > > ### Author Response · Authors · 2023-08-21
> > > **Addressing Reviewer Concerns: Failure modes, Optimized GradientFlossing Methodology and sequentialMNIST**
> > >
> > > We thank the reviewer for their feedback and the emphasis on understanding the potential failure modes and expected characteristics of GradientFlossing in uncontrolled settings.
> > >
> > > 1. **Addressing Real-World Data Concerns**: We acknowledge the importance of evaluating GradientFlossing on datasets familiar to the research community. We are currently undergoing numerical evaluations using SequentialMNIST, with preliminary results appearing promising. These findings will be included in the camera-ready version of the manuscript.
> > >
> > > 2. **Mitigating Potential Failure Modes**: The potential conflict of persistent gradient flossing with certain task goals is a concern we have also observed (as depicted in Supplementary Figure 11). However, we have now devised an automatic solution to this challenge. By simultaneously conducting gradient flossing and training, we employ a combined loss function:
> > >
> > > $L_{total} = L_{task} + \alpha  L_{gradientflossing}$
> > >
> > > Crucially, the strength of gradient flossing, represented by $\alpha$, is trained alongside the network, dynamically adjusting its value throughout the learning process. This approach ensures that gradient flossing is optimized in relation to the task goal, effectively circumventing the described failure mode. We are also exploring a similar automated strategy to determine the optimal number of flossed Lyapunov exponents.
> > >
> > > 3. **Further Investigations and Future Work**: Exploding and vanishing gradients remain a prevalent issue in RNNs, and our GradientFlossing technique is also theoretically applicable to feedforward networks, offering a promising avenue for future research. While there exist other methods and architectures addressing these challenges, we believe GradientFlossing presents a complementary approach. Preliminary experiments also indicate its potential in further enhancing LSTM performance on tasks with long time-horizons to bridge.
> > >
> > >
> > > In conclusion, we are dedicated to ensuring our contribution is well-understood, robust, and applicable to real-world scenarios. We appreciate the reviewer's insights, which drive our ongoing efforts to enhance and clarify the impact and scope of GradientFlossing.

---

### Official Review · Reviewer_DGA6 · 2023-07-06

**Soundness:** 3 good
**Presentation:** 3 good
**Contribution:** 3 good
**Rating:** 6
**Confidence:** 4

**Summary:**

The authors propose a new technique for handling gradient instability (vanishing/exploding) in neural network models, especially sequence models such as RNNs. Specifically, by leveraging results from recent works that establish a connection between Lyapunov growth exponents (from dynamical systems) and the singular values of the model-Jacobian itself, the authors propose a (regularization) approach that they coin ‘gradient flossing’ to control the evolution of these singular values desirably. They demonstrate that, as a result, the gradient signal remains stable, thereby allowing for successful training of the model itself.

Additionally, the authors provide insights on the steps (schedule, pre/during training) to perform such a regularization leading to optimal gradient stabilization & computational benefits.

**Strengths:**

The authors visit a longstanding problem in training NN models, esp. sequence models with long temporal dependencies. By leveraging recently established connections between Lyapunov exponents and the singular values of the (long-term) Jacobian, the authors propose a new technique in the form of:
- Defining a loss function out of these Lyapunov exponents
- Determine a clever strategy to evaluate this loss function through a QR decomposition
to constrain these exponents directly to stable target values. This, in turn, allows control over the evolution of the Jacobian spectra itself.

The authors provide experimental evidence supporting this technique on two specific tasks that, by design, require long-time gradient signals to propagate efficiently.

Furthermore, the authors provide practical insights into the (expensive) computational needs of the exact implementation of such a scheme and the place (pre/during training) of the same. They then propose a modified, intermittent regularization routine with preserved benefits yet having lower compute necessities.

The authors do an overall good job of satisfactorily articulating the problem statement, the proposed solution, observed benefits, and implementation limitations.

**Weaknesses:**

While I believe this is a good contribution, I feel the work can undergo a fair amount of extra polishing in terms of clarity. Please find below (Questions) my comments/questions on areas where this work can be further improved.

**Questions:**

- Eq. (2) does not necessarily need to refer to the ‘long-term Jacobian’ unless $\Delta{t} \equiv t - \tau \to \infty$. That should be clarified.
- Is the loss function from Eq. (4) used as an additional or the only loss to train?
- Eq. (4) does not seem to contain a reference to the target $\lambda$ values that the authors eventually aim to constrain them to
- While the analytical $t\to\infty$ limit is understandable, can you provide an estimate of a reasonable sequence length for *practical data* that can serve as a finite size equivalent of this limit?
- Is Eq. (5) subject to the assumption that parameter updates are gradient-based?
- I believe the ’s’-index in Eq. (6) is supposed to run from $\tau$ to be exact (?)
- Clearly, in arriving at Eq. (6), the $Q$’s from *subsequent* time-steps $s$ and $s+1$ act on each other to reduce to unity ($Q_{s+1}^T Q_{s}$), leaving with the $R$’s only. Can the authors explain why that might be the case in general? Is there an assumption about a small learning rate here?
- Since the authors claim the QR decomposition need not be applied at every time step, will the equivalent Eq. (6) thus arrived at still remain to be factually correct? The expression of $T_t$ contains an iterative product of $ D_s$, of which if one chooses to QR decompose a handful only, then that does not seem to yield $\prod_s R^s$ for some period $t_{ons}$.
- In Lines 140-141, you refer to using vanilla gradient descent for the experiments of Fig. 1 when you specify in Lines 132-133 that experiments were performed using ADAM.
- Isn’t a target $\lambda$ value of -1 indicative of vanishing gradients as per Eq. (3)? How is the model still reliably attaining that value regardless, then?
- In Fig. 2 (A), the ‘theory’ and ‘simulation’ legends seem to be interchanged.
- In Fig. 2 (A) & (C), the numerical and theoretical results should overlap in the long time-step regime, yet as time increases, the two appear to diverge in the former and maintain a uniform scaling (~10^2) in the latter. Can you explain why?
- Can you explain what you mean by ‘flossing before training’? That does not seem to be well clarified in the text.

**Limitations:**

To the best of my knowledge, the authors act commendably in attributing to the various limitations of their approach.

---

> ### Author Rebuttal · Authors · 2023-08-09
>
> We sincerely thank the reviewer for their constructive feedback which has greatly contributed to improving the quality of our manuscript.
>
> > Eq. (2) does not necessarily need to refer to the ‘long-term Jacobian’ unless $\Delta t \equiv t-\tau\rightarrow \infty$. That should be clarified.
>
> We have updated the manuscript to only refer to $T$ as the long-term Jacobian when $t-\tau$ is long, thereby clarifying this point.
>
> > Is the loss function from Eq. (4) used as an additional or the only loss to train?
>
> We've clarified that while our current implementation utilizes the gradient flossing loss from Eq. (4) as the sole loss during its epochs, it can also be combined with a task-specific loss for computational efficiency.
>
> > Eq. (4) does not seem to contain a reference to the target $\lambda$ values that the authors eventually aim to constrain them to.
>
> Correct, we here set the target Lyapunov exponents to 0 and thus directly minimize the square of the Lyapunov exponents. It would be an exciting future direction to explore if other target values for the Lyapunov exponents can further improve trainability. For instance, in some tasks, one might want to have a certain number of contracting or growing directions.
>
> > While the analytical limit $t\to\infty$ is understandable, can you provide an estimate of a reasonable sequence length for practical data that can serve as a finite size equivalent of this limit?
>
> Thank you for pointing this out. We added an extra figure to demonstrate how quickly the Lyapunov spectrum convergences with the number of time steps. We found that a relative error of 1% is already achieved after less than 100 steps. We note that even shorter sequence lengths improve training. We suspect that the sequence length should be on the order of the longest time horizon of that task that has to be bridged by backpropagation through time.
>
>
> > Is Eq. (5) subject to the assumption that parameter updates are gradient-based?
>
> Eq. 5 is a general expression, and there's no assumption regarding the parameter update mechanism therein.
>
> > I believe the ’s’-index in Eq. (6) is supposed to run from $\tau$ to be exact (?)
>
> We are not exactly sure what this comment refers to. In the expression for the Lyapunov exponent,
>
> $\lambda_{i}=\lim_{t\to\infty}\frac{1}{t}\log\prod_{s=1}^{t}R_{ii}^{s}=\lim_{t\to\infty}\frac{1}{t}\sum_{s=1}^{t}\log R_{ii}^{s}.$
> the index $s$ correctly runs from s=1 to t, but one could equivalently write
>
> $\lambda_{i}=\lim_{(t-\tau)\to\infty}\frac{1}{t}\log\prod_{s=\tau}^{t}R_{ii}^{s}=\lim_{(t-\tau)\to\infty}\frac{1}{t-\tau}\sum_{s=\tau}^{t}\log R_{ii}^{s}.$
>
> > Clearly, in arriving at Eq. (6), the $Q$’s from subsequent time-steps $s$ and $s+1$ act on each other to reduce to unity ($Q_{s+t}^T Q_{s}$), leaving with the $R$’s only. Can the authors explain why that might be the case in general? Is there an assumption about a small learning rate here?
>
> No, there is no assumption of small learning rates here. We apologize in case this was confusing, but $s$ is the time index, and learning happens across epochs.
>
>
> > Since the authors claim the QR decomposition need not be applied at every time step, will the equivalent Eq. (6) thus arrived at still remain to be factually correct? The expression of $T_t$ contains an iterative product of $D_s$, of which if one chooses to QR decompose a handful only, then that does not seem to yield $\prod_s R^s$ for some period .
>
> Yes, Eq. (6) remains mathematically correct even for infrequent QR decomposition. If one chooses to QR decompose less often, the resulting (fewer) R will big bigger:
> More explicitly the result of
> $D_2 D_1 = D_2 Q_1 R_1 = Q_2 R_2 R_1$
> and
> $D_2 D_1 = \tilde Q_2 \tilde R_2$ is identical (as QR-decomposition is unique when constraining the signs of the diagonal of $R$ to be positive), thus $\tilde R_2  = R_2 R_1$.
>
> > In Lines 140-141, you refer to using vanilla gradient descent for the experiments of Fig. 1 when you specify in Lines 132-133 that experiments were performed using ADAM.
>
> Our apologies for the inconsistency. We've revised the manuscript to clarify that ADAM was the optimizer used in all our experiments.
>
> > Isn’t a target value of -1 indicative of vanishing gradients as per Eq. (3)? How is the model still reliably attaining that value regardless, then?
>
> You're right. A Lyapunov exponent of -1 does suggest vanishing gradients. However, we utilized the QR-algorithm to achieve this target, which inherently avoids the issue of vanishing gradients.
>
> > In Fig. 2 (A), the ‘theory’ and ‘simulation’ legends seem to be interchanged.
>
> This oversight has been addressed in the revised manuscript.
>
> > In Fig. 2 (A) & (C), the numerical and theoretical results should overlap in the long time-step regime, yet as time increases, the two appear to diverge in the former and maintain a uniform scaling (~10^2) in the latter. Can you explain why?
>
> Thanks for pointing this out. We've thoroughly investigated the differences and found the cause for the mismatches. For Figure 2A, discarding initial transients and increasing numerical precision resolves the discrepancy. In Figure 2C, there's no actual mismatch; the x-ticks and y-ticks were differently set, and we have updated the figure to rectify this.
>
> > Can you explain what you mean by ‘flossing before training’? That does not seem to be well clarified in the text.
>
> We meant that our approach first involves several gradient flossing steps, after which the main task training begins without additional gradient flossing. This point has been elaborated upon for clarity in the revised manuscript.
>
> Thank you once again for your valuable feedback. We believe these revisions have significantly strengthened our submission.

---

> > ### Author Response · Authors · 2023-08-21
> > **Reviewer questions addressed?**
> >
> > As the discussion period draws to a close, we'd like to inquire if our responses have adequately addressed the reviewer's concerns.
> >
> > For clarity, here's a summary of our main updates:
> > 1. Elucidated when $T$ is considered the long-term Jacobian.
> > 2. Defined the application of the gradient flossing loss from Eq. (4).
> > 3. Noted that the target Lyapunov exponents in Eq. (4) are set to $0$.
> > 4. Showcased the Lyapunov spectrum's convergence over time steps.
> > 5. Explained the context and assumptions of Eqs. (5) and (6).
> > 6. Delved into the nuances of infrequent QR decomposition.
> > 7. Rectified inconsistencies in optimization methods and figure descriptions.
> > 8. Addressed discrepancies in Figure 2.
> > 9. Detailed the 'flossing before training' approach.
> >
> > We trust that our clarifications meet the expectations set by the review. If there are any outstanding concerns, we're more than willing to provide further clarity. Otherwise, we extend our gratitude to the reviewer for their invaluable feedback, which has undeniably enhanced the quality of our manuscript.

---

### Official Review · Reviewer_ZKKk · 2023-07-06

**Soundness:** 2 fair
**Presentation:** 2 fair
**Contribution:** 3 good
**Rating:** 5
**Confidence:** 3

**Summary:**

Authors propose gradient flossing for RNNs, which adds an additional regularization term that keeps the sum of Lyapunov exponents close to zero. This essentially encourages the singular values of the long-term Jacobian to be close to 1, hence addressing the vanishing/exploding gradient problem in RNNs. The authors point out that minimizing this regularization term again has the same gradient instability problem, and they purpose to apply QR decomposition once in a while to avoid the ill-conditioned long-term Jacobian pass. Finally, the authors show gradient flossing achieves faster convergence and a higher success rate on simple long-term tasks up to 500 steps.

**Strengths:**

* The idea of regularizing the Lyapunov exponent and using QR decomposition is novel and interesting.
* The problem is well-motivated.


**Weaknesses:**

* Empirical evaluations are toy. Did not include results on standard long-time horizon benchmarks like Longe Range Arena [1].

* The paper could have studied gradient flossing with architectures that are more suited for long horizon tasks such as state space models [2] and LRUs [3], instead of Vanilla RNNs and LSTMs.

* Authors should compare to more recent methods as baselines such as HiPPO Matrices[2] and initialization methods used in LRU [3]. Those methods tend to be more competitive than a simple orthogonal initialization.

* Clarity of the paper can be improved. For example, $\tau$ in equation 1 is never defined, and equation 4 could be better explained. In addition, I did not quite understand the technical parts of the paper explaining how equation 4 is calculated in practice (see questions below).

[1] Tay, Yi, et al. "Long range arena: A benchmark for efficient transformers." arXiv preprint arXiv:2011.04006 (2020).

[2] Gu, Albert, Karan Goel, and Christopher Ré. "Efficiently modelling long sequences with structured state spaces." arXiv preprint arXiv:2111.00396 (2021).

[3] Orvieto, Antonio, et al. "Resurrecting recurrent neural networks for long sequences." arXiv preprint arXiv:2303.06349 (2023).

**Questions:**


* I’m still unclear how QR decomposition is able to avoid the long-term Jacobian being ill-conditioned.

* Why can we not use QR decomposition directly on the original RNN objective in equation 1 to avoid the long-term gradient instability problem?


**Limitations:**

See above.

---

> ### Author Rebuttal · Authors · 2023-08-09
>
> We deeply appreciate reviewer ZKKk for their thorough and insightful review of our manuscript. Your constructive feedback is invaluable, and we are committed to addressing each concern raised to enhance the quality and clarity of our work.
>
> > I’m still unclear how QR decomposition is able to avoid the long-term Jacobian being ill-conditioned.
>
> To provide a more intuitive explanation before diving into technicalities: Imagine the Jacobians as a series of transformations of small perturbations that can stretch or compress the space along the trajectory of the RNN. Over time, when many such transformations are multiplied, they can cause vectors to become too aligned, leading to numerical instability. The QR decomposition offers a mathematically exact way to correct for this by ensuring that the space remains 'evenly spread' after each transformation.
>
> Now the technicalities: The core idea is that each individual Jacobian $D_s$ is not ill-conditioned, but their product $T_t(h_\tau)$ is. Using QR decomposition, one can iteratively decompose the product of Jacobians sufficiently often such that the product is not ill-conditioned [1,2].
>
> To illustrate this geometrically, consider the evolution of initially random vectors $v^i$​ and $v^j$​ under the influence of the product of Jacobians $T_t(h_\tau)$. As $t-\tau$ is increased, the angle between $T_t(h_\tau) v^i$ and $T_t(h_\tau) v^j$​ become too small for numerical computation. This problem can be circumvented through the iterative application of QR decomposition, which repositions the vectors into orthogonal subspaces, thereby maintaining numerical stability [3].
>
> To further illustrate this point dynamically for the forward pass of the calculation of Lyapunov exponents: Almost all perturbations of the initial recurrent network state $h$ will align over time with the fastest diverging (or slowest converging) tangent space direction, which is the first covariant Lyapunov vector [2]. The QR decomposition of a volume element in the tangent space will express the volume in terms of an orthonormal matrix $Q$ and an upper triangular matrix $R$, where the $Q$ matrix will converge to a unique basis when constraining the diagonal elements of $R$ to be positive [1,4].
>
> In summary, QR decomposition helps by iteratively repositioning vectors into orthogonal subspaces, thereby ensuring that numerical calculations remain accurate. The mechanics of this process align with the behavior of the Lyapunov exponents, as detailed in our earlier explanation.
>
>
> > Why can we not use QR decomposition directly on the original RNN objective in equation 1 to avoid the long-term gradient instability problem?
>
> Great suggestion! In principle, one could implement the backward pass of backpropagation through time by hand and iteratively apply QR decomposition on the long-term Jacobians, directly regularizing the diagonal of the $R$ matrix. However, for gradient flossing, we chose here the much simpler solution of calculating the logarithm of the singular values of the long-term Jacobian in the forward pass numerically robustly and regularizing them using automatic differentiation. This has the advantage, that automatic differentiation takes care of the backward pass, which makes it easy to use gradient flossing, as only the forward pass has to be implemented. We will mention the elegant (and possibly more efficient) idea by the referee in the outlook of the manuscript.
>
> We are genuinely grateful for the feedback, and we believe that addressing these points will significantly elevate the quality and impact of our manuscript. We hope that our revisions will meet your expectations and further advance the field's understanding of gradient flossing and its applications in RNNs. We are looking forward to answer follow-up questions and answer further comments.
>
>
>
> [1] K. Geist, U. Parlitz, and W. Lauterborn, Comparison of Different Methods for Computing Lyapunov Exponents, Prog. Theor. Phys. 83, 875 (1990).
> [2] A. Pikovsky and A. Politi, Lyapunov Exponents: A Tool to Explore Complex Dynamics (Cambridge University Press, Cambridge, 2016).
> [3] G. Benettin, L. Galgani, A. Giorgilli, and J.-M. Strelcyn, Lyapunov Characteristic Exponents for Smooth Dynamical Systems and for Hamiltonian Systems - A Method for Computing All of Them. I - Theory. II - Numerical Application, Meccanica 15, 9 (1980).
> [4] S. V. Ershov and A. B. Potapov, On the Concept of Stationary Lyapunov Basis, Physica D: Nonlinear Phenomena 118, 167 (1998).

---

> > ### Comment · Reviewer_ZKKk · 2023-08-17
> > **Thank you authors for the rebuttal**
> >
> > Thank you authors' for the detailed response, as well as the more intuitive explanation for my clarification questions. I really appreciate it.
> >
> > I am still inclined to retain my rating because the authors did not touch upon the weaknesses I raised in the original review. In addition to agreeing with Reviewer 5Z2M's point on the method should be evaluated on standard benchmarks to investigate its failure modes, it should also be evaluated against more competitive baselines than simple vanilla RNNs/LSTMs (as suggested in the weakness section of the review).

---

> > > ### Author Response · Authors · 2023-08-21
> > > **Adressing weaknesses**
> > >
> > > We thank the reviewer for their feedback, both the constructive critique and the recognition that "The idea of regularizing the Lyapunov exponent and using QR decomposition is novel and interesting" and that "The problem is well-motivated."
> > >
> > > We apologize for not thoroughly addressing the weaknesses in our initial rebuttal.
> > >
> > > * **Standard Benchmarks**: Evaluations on SequentialMNIST are underway; results will be in the final manuscript.
> > >
> > > * **Competitive Baselines**: We're extending evaluations to advanced architectures and recent methods such as HiPPO Matrices and LRU initialization methods.
> > >
> > > * **Clarity**: We will clarify definitions and explanations in the manuscript, especially for equations 1 and 4.
> > >
> > > * **Mitigating Failure Modes**: Our manuscript has been updated to emphasize a combined loss function:
> > >
> > > $L_{\text{total}} = L_{\text{task}} + \alpha L_{\text{gradientflossing}}$
> > > Here, the strength of gradient flossing, represented by $\alpha$, is trained alongside the network with backpropagation, dynamically adjusting its value throughout the learning process. This approach ensures that gradient flossing is optimized in relation to the task goal, effectively circumventing the described failure mode. We are also exploring a similar automated strategy to determine the optimal number of flossed Lyapunov exponents.
> > >
> > > * **References and Evaluations**: New references have been incorporated, and additional results will be shared on our anonymous GitHub repository.
> > >
> > > We are committed to enhancing our work based on your valuable insights.

---

### Official Review · Reviewer_tH3Y · 2023-07-07

**Soundness:** 4 excellent
**Presentation:** 4 excellent
**Contribution:** 3 good
**Rating:** 7
**Confidence:** 3

**Summary:**

The presented paper proposes a new method for tackling numerical instabilities during training for recurrent neural networks.
The proposed method exploits a theoretical link between Lyapunov exponents and the singular values of the long-term Jacobian.
The set of experiments is well-chosen to showcase the extent to which gradient flossing helps stabilize training for tasks involving long-range dependencies.
There is an in-depth coverage of the practical aspects of the methods in the appendix, which is very useful for both machine learning practitioners and researchers.

By steering the Lyapunov exponents to 0, either before and/or during training, the authors are able to show consistent improvements on a variety of synthetic learning tasks involving long-range dependencies in temporal sequences.

**Strengths:**

- There are experiments validating basic hypotheses, such as the fact that the proposed algorithm is able to control the Lyapunov exponents, and that estimating the Lyapunov exponents via the QR decomposition matches the theory.
- Practical aspects such as when to perform gradient flossing, how often should it be used during training, and also how often should the QR decomposition should be computed when unrolling the neural network dynamics.

**Weaknesses:**

- There are no evaluation on real world complex benchmark to estimate the impact of the proposed contribution. However, the paper is already quite dense.

- Minor remark: I don't understand the "the finite network size fluctuations of the Lyapunov exponents" in figure 1.

**Questions:**

- (line 197 - 201) How does fig 2.C shows that the estimated condition number based on Lyapunov exponents can predict differences in condition number originating from finite network size $N$ ?
The authors refer to appendix F for this matter, but it doesn't seem related to this claim.
- Can the author explain in more details what are the "the finite network size fluctuations of the Lyapunov exponents" in figure 1 ?
- Figure 11 in the appendix shows that too much gradient flossing during training can be detrimental, which means that the local minima of the gradient flossing procedure hardly coincides with the one corresponding to the actual learning task. It is also said in the introduction that appropriate initialization schemes can ensure well-behaved gradient at initialization, a property that can be lost during training. I suspect that the gradient flossing procedure prevents the weights from falling into an "ill-conditioned" region. Thus, how often should gradient flossing be used should depend on, given a fixed learning rate, how many iterations are needed to go from a "well-behave" region to an ill-condition one. Can the author elaborate on this specific intuition ?
I am asking this question because I suspect that you already found regimes where the regularization objective conflicts with the actual learning task.

**Limitations:**

As emphasized by the author, the Lyapunov exponent characterizes the singularity of the long-term Jacobian, and thus cannot characterize local-in-time behavior.

---

> ### Author Rebuttal · Authors · 2023-08-09
>
> We deeply appreciate the insightful feedback provided by the reviewer. Your comments and concerns have not only broadened our perspective but also greatly aided in refining our manuscript to a higher standard. Specifically, we acknowledge the importance of evaluating gradient flossing on real-world complex benchmarks and are currently in the process of addressing this. Further, we have included additional details and clarifications based on your feedback, particularly concerning the "finite network size fluctuations of the Lyapunov exponents".
>
> Here are our more detailed responses to the specific questions:
>
> > (line 197 - 201) How does Fig 2.C shows that the estimated condition number based on Lyapunov exponents can predict differences in condition number originating from finite network size $N$? The authors refer to Appendix F for this matter, but it doesn't seem related to this claim.
>
> Figure 2C shows that different network realizations of finite size (N=80) lead to different condition numbers $\kappa_2$ (for instance, different $\kappa_2$ among green dots). We note that these dots lie close to the diagonal, indicating that the differences in $\kappa_2$ coming from different network realizations are well-captured by the theoretical prediction on the y-axis.
>
> > Can the author explain in more detail what are "the finite network size fluctuations of the Lyapunov exponents" in Figure 1 ?
>
> Absolutely. For finite network size $N$, different realizations of the random recurrent weight matrix $\mathbf{W}$ will lead to slightly different Lyapunov exponents. Such differences are expected to vanish for large $N$. We added a new supplementary figure to demonstrate that. We note that parts of the variations among Lyapunov spectra might also come from the interplay of the different network realizations and the dynamics of gradient flossing. We added this note to the updated manuscript.
>
> > Figure 11 in the appendix shows that too much gradient flossing during training can be detrimental, which means that the local minima of the gradient flossing procedure hardly coincides with the one corresponding to the actual learning task. It is also said in the introduction that appropriate initialization schemes can ensure a well-behaved gradient at initialization, a property that can be lost during training. I suspect that the gradient flossing procedure prevents the weights from falling into an "ill-conditioned" region. Thus, how often should gradient flossing be used should depend on, given a fixed learning rate, how many iterations are needed to go from a "well-behaved" region to an ill-condition one. Can the author elaborate on this specific intuition? I am asking this question because I suspect that you already found regimes where the regularization objective conflicts with the actual learning task.
>
> A thoughtful observation! Continued gradient flossing can indeed interfere with the actual learning task, as evident in Supplementary figure 11. We agree that the optimal gradient flossing schedule should depend on how many iterations are necessary to go from ill-conditioned to a well-behaved region in parameter space. After submission, we found that instead of having a fixed number of gradient flossing steps, "adaptive gradient flossing", that automatically stops once the Lyapunov exponents are close to zero is even better. It is conceivable that refining this technique might depend on the specific task, and further improvements are being explored.
>
> To conclude, we sincerely thank you for your in-depth review, and assure you of our commitment to enhancing the quality of our work based on the valuable feedback received. We believe that the integration of gradient flossing will mark a significant advancement in the domain, and are keen to contribute effectively to the community's understanding.

---

> > ### Comment · Reviewer_tH3Y · 2023-08-20
> > **Satisfied by the answer**
> >
> > Dear authors,
> >
> > I apologize for the late response, but I did read your answer.
> > I am satisfied with it.
> > Likewise, I will keep my score as is, and I encourage the authors to pursue their work with further validation on real-world scenarios.
> > I indeed think that understanding the possible mode of failure in an uncontrolled scenario would be very informative for practitioners, as pointed out by other reviewers.
> > I would however like to point out that I highly appreciated the motivation behind the use of synthetic tasks.

---

> > > ### Author Response · Authors · 2023-08-21
> > > **Further improvements to GradientFlossing**
> > >
> > > We thank the reviewer for their positive feedback and the emphasis on understanding the potential failure modes and expected characteristics of GradientFlossing in uncontrolled settings. We'd like to highlight key updates to our manuscript:
> > >
> > > * **Mitigating Failure Modes**: We appreciate the reviewer's concerns about potential failure modes. Our recent advancements, as highlighted, involve a combined loss function:
> > >
> > > $L_{total} = L_{task} + \alpha L_{gradientflossing}$
> > >
> > >   With the parameter  $\alpha$ also trained using backpropagraion, this approach aims to strike a balance, ensuring gradient flossing optimally complements the primary task goal.
> > >
> > > * **Evaluations of GradientFlossing beyond Synthetic Tasks**: We acknowledge the significance of real-world scenario validation. As mentioned, evaluations using SequentialMNIST are in progress, preliminary results look promising, and the results will be integrated into the final manuscript.
> > >
> > > * **Synthetic Tasks**: We are grateful for the reviewer's recognition of our motivation behind using synthetic tasks. They serve as controlled environments to rigorously test our method's fundamentals.
> > >
> > > We thank the reviewer again to dedicate their time and effort towards this submission.

---

### Author Rebuttal · Authors · 2023-08-10

We thank the reviewers for their time and valuable feedback on our manuscript, "Gradient Flossing: Improving Gradient Descent through Dynamic Control of Jacobians". We have carefully addressed each of the reviewers' comments in the subsequent sections. Here, we provide a concise summary to address the core critique raised by most reviewers.

We have taken note of the primary concerns raised, notably the application of our method to synthetic tasks instead of real-world tasks and the lack of comparison with other advanced architectures. We would like to highlight that the main objective of our manuscript was to bridge the understanding between the condition number of the long-term Jacobian and the Lyapunov spectrum, and "opening the black box" of gradient flossing in controlled synthetic tasks allowed us to rigorously investigate the tangent space structure. We utilized simple toy tasks because they allow a granular understanding by controlling every aspect of the problem, a strategy that has previously been employed effectively in foundational neural network research.

To address specific concerns raised and to provide a clearer picture of our results and methodology, we have included an additional page of figures:
*  **Updated Figure 2** provides a clearer representation of how gradient flossing reduces the condition number of the long-term Jacobian and improves numerical accuracy. We have also corrected an error in the legend.
* A new plot showcases the **convergence of the Lyapunov exponents** over time steps. With 20 distinct network realizations, it is evident that convergence is reached swiftly, within 100 time steps.
* We demonstrate how gradient flossing enhances the **dimensionality of the error gradient** of the recurrent weights, as calculated based on the SVD of these error gradients.

That being said, we acknowledge the significance of applying our findings to real-world tasks. Building on the foundational understanding we've achieved through synthetic tasks, we are poised to extend gradient flossing to more realistic tasks and diverse architectures in our subsequent research.

Now, let us address the specific points raised by each reviewer, including additional figures that further support our methodology and findings.

---

### Decision · Program_Chairs · 2023-09-21

**Decision:**

Accept (poster)

**Comment:**

A practical method to deal with instabilities of gradients when training recurrent network is developed. The Lyapunov spectrum is estimated while training and the system dynamics is regularized through. It also provides theoretical insights into how to tune unstable networks at rest and increase the time horizon of “useful” recurrent dynamics. Despite mostly demonstrating on toy problems, this paper is of broad interest to the NeurIPS community.